# GRAPH UNLEARNING VIA RECONSTRUCTION
# – A RANGE-NULL SPACE DECOMPOSITION APPROACH

## ABSTRACT

Graph unlearning is a machine unlearning technique tailored to graph neural networks (GNNs) to remove nodes or edges from the training graph. Conventional methods such as retraining is highly inefficient, while influence function-based approaches merely work on minor removal, like 10% or less of the graph edges. To resolve the problems, we reverse the aggregation process in GNN training by modeling the interaction between unlearned nodes and their neighbors. Given one unlearned node, its embedding is roughly disassembled and assigned to its neighbours by reconstruction, and then removed from its neighbours by embedding modification. We also introduce range-null space decomposition to rectify the raw estimation of the interaction with theoretical support. Experimental results on multiple representative datasets and GNN models demonstrate the efficiency of at least $40\times$ acceleration compared with retraining and superior unlearning utility, efficacy, and privacy of our proposed approach compared with other methods.

## 1 INTRODUCTION

Machine unlearning removes the impact of some training data from the machine learning models upon requests (Cao and Yang, 2015). It is essential in many critical scenarios, such as the enforcement of laws concerning the protection of the right to be forgotten (Kwak et al., 2017; Pardau, 2018; Regulation, 2018), and the demands for model providers to revoke the negative effect of poisoned data (Rubinstein et al., 2009; Zhang et al., 2022), wrongly annotated data (Pang et al., 2021), or out-of-date data (Wang et al., 2022). The problem is particularly hard for graph data, since retraining a graph neural network (GNN) from scratch to delete a node incurs exorbitant computational overhead. Efficient methods (Wu et al., 2023a) depending on influence function have been explored, but the influence estimation of the unlearned node to the GNN is far from being exact.

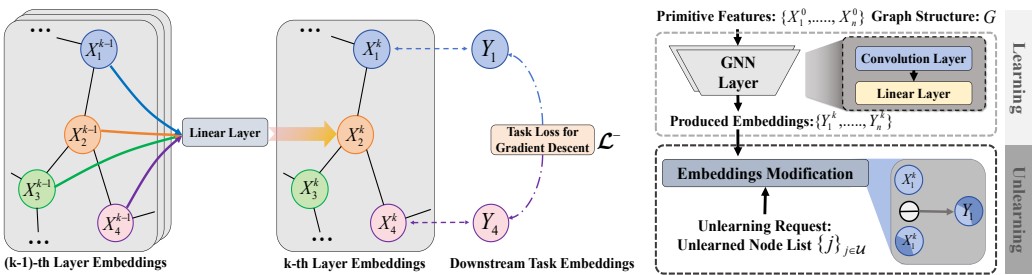

Figure 1: Brief description of graph models.     Figure 2: Proposed unlearning diagram.

To be specific, the *first* challenge of graph unlearning lies in the complicated entanglement of node features woven in the GNN. Node embedding is obtained by aggregation and transformation of the representations of its neighboring nodes, as shown in Fig. 1. Among the three types of graph unlearning tasks *edge unlearning*, *feature unlearning*, and *node unlearning*, this challenge is even more difficult to overcome on the last type of task as the node-related local neighborhood information — including node features and adjacent edges — is fully removed. Graph influence function-based methods, represented by GIF (Wu et al., 2023a), could be affected for their influence approximation under node unlearning circumstances. Its performance drops evidently, where more disturbances lead

to more error accumulation. Learning-based methods rely on ideal assumptions (Cheng et al., 2023) and adaptive selection (Li et al., 2024), while other methods like Cong and Mahdavi (2023) are more specialized in design, both types of which can't handle various complicated entanglement.

The *second* challenge to be resolved is the repetitive computation overhead of unlearning under multiple requests. The unlearning requests could be many in application, while mostly previous work considers efficiency only once under full parameters re-estimation. SISA-based (Sharded, Isolated, Sliced, and Aggregated) methods (Bourtoule et al., 2021; Chen et al., 2022a;b) propose to conduct a reasonable division of graph data into disjoint shards and retrain its sub-models, then aggregate to eliminate repetitive computation overhead. However, division incurs additional cost, and the destruction of the graph structure leads to performance degradation. GNNdelete (Cheng et al., 2023) proposes to introduce a learnable deletion operator for each layer, while other parameters are to be frozen. The deletion operators would incur more overhead for a deeper GNN. Additionally, it works under Granger causality; thus, it couldn't be easily generalized.

We propose a method that can resolve both challenges. For the *first* challenge, we disassemble complicated entanglement into a general node-wise interaction, which can be learned by reconstructing the embeddings of the unlearned nodes. With the deepening of GNN layers, node embedding is reduced in dimension and compressed in information. Reconstruction of node embeddings from output to the input constitutes an *under-determined problem*, thus more challenging. The issue is resolved by range-null space decomposition with verifiably restricted $L_2$-norm error.

For the *second* challenge, we choose to modify output node embeddings to nullify the impact of the unlearned nodes $\mathcal{U}$ as Fig. 2, to mimic that of trained from scratch, instead of altering the full GNN parameters. Various unlearning requests could be resolved on the shared embeddings of the original. The key observation is that, contrary to the aggregation of message passing in GNN learning, the unlearning should deduct the unlearned nodes' influence within their neighbors. An unrolling of the gradient descent is also performed for the downstream task with a local search loss.

Highlights of our contribution are as follows. (**1**) The proposed method decouples the learning and unlearning by directly modifying shared original embeddings to satisfy diverse unlearning requirements. This approach eliminates the necessity for re-estimating or retaining multiple full-parameter models, offering a more efficient and conceptually coherent solution. (**2**) The reconstruction module takes the outputs of the original as input; it takes effects without adaptation to specific model structures. The range-null space decomposition rectifies the raw estimation with theoretically restricted $L_2$-norm error. (**3**) Extensive experiments on three real-world graph datasets and four state-of-the-art GNN models have illustrated the superior unlearning efficiency of acceleration of $40\times$ to $88\times$ compared with retraining. The unlearned model utility is closer to retraining compared to other works. Its efficacy is also remarkable, and its privacy could be verified under a membership inference attack.

## 2 RELATED WORK

### 2.1 MACHINE UNLEARNING

Machine unlearning aims to eliminate the influence of a subset of the training data from the trained model for privacy, security, or utility concerns. Ever since Cao and Yang (2015) first introduced the concept, several methods have been proposed to address the unlearning tasks, which can be classified into exact (Ginart et al., 2019; Karasuyama and Takeuchi, 2010; Bourtoule et al., 2021) or approximate unlearning (Koh and Liang, 2017; Guo et al., 2020; Izzo et al., 2021).

The former methods typically involve re-training from scratch so that the unlearned model is identical to the one trained without the unlearned samples. Such methods include SISA (Bourtoule et al., 2021) that partitions the data, separately trains a set of constituent submodels, and aggregates them into one model. Merely the submodels affected by the unlearned samples are retrained for aggregation.

By model fine-tuning, the approximate unlearning is more efficient than exact unlearning. For example, Guo et al. (2020) proposes removing the impact of the unlearned sample from the model by using the influence function (Koh and Liang, 2017) for estimation. Unrolling SGD (Thudi et al., 2022) introduces a regularizer to reduce the 'verification error,' which is an approximation to the distance between the unlearned model and a retrained-from-scratch model. Langevin unlearning (Chien et al., 2024) leverages the Langevin dynamic analysis for unlearning. Details in appendix A.1.

## 2.2 GRAPH UNLEARNING

Graph unlearning is tailored for GNNs trained with graph data, which could be divided into (1) The Shards-based method: GraphEraser (Chen et al., 2022b) and GUIDE (Wang et al., 2023) extend the shards-based idea to graph-structured data, which offers partition methods to preserve the structural information and also designs a weighted aggregation for inference. Moreover, GraphRevoker (Zhang, 2024) utilized a property-aware sharding method and contrastive sub-model aggregation for efficient partial retraining and inference. They rely too much on a reasonable division of graph data into disjoint shards and these sub-models' costly retraining. Its inference cost is also higher as it requires results aggregation from all sub-models; (2) The IF-based method: CGU (Chien et al.), GIF (Wu et al., 2023a), CEU (Wu et al., 2023b), IDEA (Dong et al., 2024), and GST (Pan et al., 2023) leverage rigorous mathematical formulations to quantify the impact of data removal on model, allowing for efficient model updates under the Lipschitz continuous condition and loss convergence condition. These methods work under the unlearning requests that minorly change the graph structure, like 10% of graph edges. Its performance drops for node unlearning; (3) Learning-based method: GNNDelete (Cheng et al., 2023) bounding edge prediction through a deletion operator under Granger causality, and MEGU (Li et al., 2024) achieved effective and general graph unlearning through a mutual evolution design with adaptive HIN set selection; (4) Others: Projector (Cong and Mahdavi, 2023) provides closed-form solutions with theoretical guarantees but is more specialized in design. It couldn't be generalized easily. Details in appendix A.2.

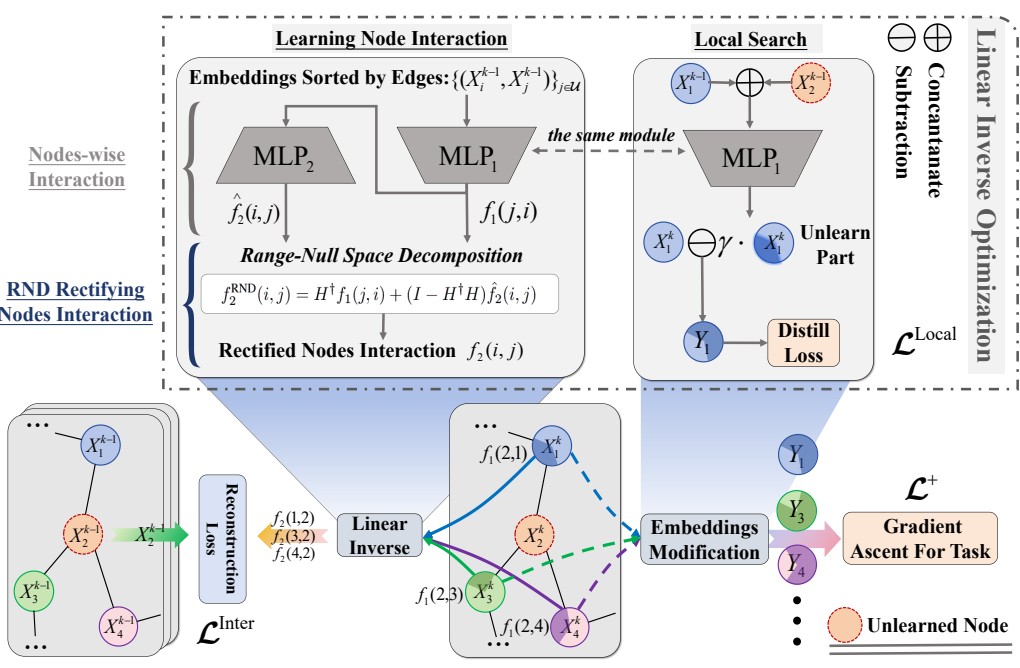

Figure 3: The illustration of our framework. Given node 2 as an example for unlearning target, its neighbor's embeddings $X_i^k$ are modified to $Y_i$ as unlearned embeddings. The $f_1$ is trained to measure the interactions between nodes by reconstructing the $(k-1)$-th layer embeddings by linear inverse optimization. Range-Null Space Decomposition is employed for $f_1$'s $(k-1)$-th layer correspondence $f_2$ in $\mathcal{L}^{\text{Inter}}$. The embeddings modification is implemented by the local search with $f_1$'s interactions subtractions through $\mathcal{L}^{\text{Local}}$. The specific unlearning also involves the gradient ascent terms $\mathcal{L}^+$.

## 3 LEARNING INTERACTION VIA NODE EMBEDDING RECONSTRUCTION

We turn the problem of graph unlearning into embedding transformation: eliminating the impact of the unlearned nodes from the retained ones so that the new embeddings are equivalent to those trained from scratch without the unlearned nodes. To achieve that, we first investigate how two nodes interact with each other on a graph.

**Modeling node-wise interaction.**    Following the forward process of GNN, the embedding $\boldsymbol{h}_{e_i}^k$ of node $e_i$ in the $k$-th layer is obtained by the $(k-1)$-th layer node embeddings as:

$$\boldsymbol{h}_{e_i}^k = \sigma \left( \sum_{e_j \in \mathcal{N}_{e_i} \cup \{e_i\}} \alpha_{i,j} W^k \boldsymbol{h}_{e_j}^{k-1} \right), \tag{1}$$

where $\alpha_{i,j}$ represents the attention coefficient between nodes, $W^k$ is the transformation matrix of the $k$-th layer, and $\sigma$ is the activation function. The embedding of the $k$-th is aggregated from those of the $(k-1)$-th, then, $W^k$ projects the $(k-1)$-th embeddings of nodes onto the $k$-th dimension.

We design $f_1(j, i)$ to depict the influence of the unlearned node $e_j$ to the retained node $e_i$ at the $k$-th layer, given their $(k-1)$-th layer embeddings:

$$f_1(j, i) = MLP_1(\boldsymbol{h}_{e_j}^{k-1}, \boldsymbol{h}_{e_i}^{k-1}). \tag{2}$$

The $f_1(\cdot)$ projects the pair of embeddings of $e_i, e_j$ onto the $k$-th layer embedding space to represent $e_j$'s impact to $e_i$. An MLP is utilized to represent the interaction. The goal is to remove the influence of the unlearned nodes within its neighbour $\mathcal{N}_{e_i}$ from the $k$-th layer embedding of $e_i$:

$$\tilde{\boldsymbol{h}}_{e_i}^k = \boldsymbol{h}_{e_i}^k - \sum_{j \in \mathcal{U} \cap \mathcal{N}_{e_i}} f_1(j, i). \tag{3}$$

Thus $\tilde{\boldsymbol{h}}_{e_i}^k$ denotes the $k$-th layer embedding if $\mathcal{U}$ is removed.

**Remark.** According to Eq. 1, the representation of the $k$-th layer is aggregated from the $(k-1)$-th layer embedding. Imitating how GNN aggregates the message, we form the nodes interaction $f_1$ in a similar way – MLP plays the roles of the linear layer $W^k$ and activation function $\sigma$. The attention for aggregation $\alpha_{i,j}$ could be figured out by embeddings $\boldsymbol{h}_{e_i}^{k-1}$ and $\boldsymbol{h}_{e_j}^{k-1}$. Therefore, it is adequate to take MLP as a function and employ $\boldsymbol{h}_{e_i}^{k-1}$ and $\boldsymbol{h}_{e_j}^{k-1}$ as input to represent the general and complex interaction between nodes, covering various types of GNNs.

**Embeddings reconstruction loss.**    To learn the nodes interaction, we adopt a second MLP $f_2(\cdot)$ to reconstruct the $(k-1)$-th layer embeddings from $f_1(\cdot)$. The $f_2(i, j)$(rectified by $f_2^{\text{RND}}$ in Sec. 4), takes $f_1(j, i)$ as the input and reverses the direction of nodes interaction, i.e., representing how much information $e_i$ passes to $e_j$ at $(k-1)$-layer:

$$f_2(i, j) = MLP_2(f_1(j, i)). \tag{4}$$

Then we aggregate the information passing from neighboring nodes to the unlearned node $e_m$ by $\sum_i f_2(i, m)$ and reconstruct its embedding at the $(k-1)$-th layer by distillation with the true value:

$$\mathcal{L}^{\text{Inter}} = \frac{1}{|m|} \sum_{m \in \mathcal{U}} KL \left( \text{Norm}[\boldsymbol{h}_{e_m}^{k-1}], \text{Norm}[\sum_i f_2(i, m)] \right) \tag{5}$$

where $\text{Norm}[\cdot]$ indicates normalization like $\text{Softmax}(\cdot)$, $KL(\cdot)$ represents KL divergence. The $\mathcal{L}^{\text{Inter}}$ is minimized over the two MLPs in reconstructing the $(k-1)$-th layer unlearned nodes embeddings.

**Why reconstructing the $(k-1)$-th layer?**    $\boldsymbol{h}_{e_i}^k$ is obtained by $\boldsymbol{h}_{e_i}^{k-1}$ with its neighbour's embeddings $\boldsymbol{h}_{e_j}^{k-1}$ while $\boldsymbol{h}_{e_j}^k$ is obtained by $\boldsymbol{h}_{e_j}^{k-1}$ and $\boldsymbol{h}_{e_i}^{k-1}$. **(1)** Reconstructing $(k-1)$-th layer embeddings constitutes a reverse correspondence with GNN forward propagation. Neighbours of one unlearned node would minus the disassembled reconstruction target, respectively, to remove its impact. $\boldsymbol{h}_{e_j}^k$ as reconstruction target leads to excessive unlearning cause the disassembled part they minus includes $\boldsymbol{h}_{e_i}^{k-1}$ that shouldn't be forgotten. **(2)** On the other hand, reconstructing former layers before $(k-1)$-th is challenging and computationally inefficient. Considering the under-determined problem in recovering higher-dimensional features from low, the $(k-1)$-th layer embeddings are designed as a reconstruction target to balance the amount of information to be learned and that to be forgotten.

## 4    RANGE-NULL SPACE DECOMPOSITION FOR RECTIFYING NODES INTERACTION

The key problem of unlearning is the absence of targeted embedding distributions that can be fitted without retraining. We resolve it from minor unlearning proportion, where the embedding distribution is near the original one. Then, the unlearning scheme is extended and covers the general case.

**Establishing the linear inverse constraint between $f_1$ and $f_2$.** Constructing $f_2$ from $f_1$ by Eq. 4 is intrinsically hard due to the information loss in dimension reduction. To overcome that, we employ the *range-null space decomposition* for rectifying the inaccurate estimation. The technique projects the representation of a vector onto the null space (i.e., $I - H^\dagger H$ term) and the range space (i.e., $H^\dagger$ term), combining both of which could give a fine estimation of the linear inverse constrained data.

The node interaction is designed to reverse the process of GNN message passing. Since the message passing between nodes are instantiated by linear transformation and non-linear activation (Eq. 1), its reverse process should be represented in a similar way. The linear reverse constraint between $f_1$ and $f_2$ could be established by a linear degenerate operator $H$ and an error term $z$ for non-linear factor:

$$f_1(j,i) - z = H \cdot f_2(j,i), \tag{6}$$

where $H$ aligned the dimension of $f_2(j,i)$ and $f_1(j,i)$, which is exactly the embedding space of the $(k-1)$-th layer to that of the $k$-th. The learned $W^k$ of the targeted GNN plays the same role; therefore, $H$ is indicated by it. We apply range-null space decomposition to estimate $f_2$ based on $f_1$, where term $z$ is eliminated to facilitate a straightforward discussion of *consistency* and *realness*.

$$f_2^{\text{RND}}(i,j) = H^\dagger f_1(j,i) + (I - H^\dagger H)f_2(i,j) \tag{7}$$

where $f_2(i,j)$ is the preliminary estimation from Eq. 4. $f_2^{\text{RND}}(i,j)$ is derived by $f_2(i,j)$ and $H^\dagger$ – a *generalized inverse matrix* of $H$ satisfying $HH^\dagger H \equiv H$. Lemma 1 verify its **consistency**:

**Lemma 1** $f_2^{RND}(i,j)$ and $f_2(i,j)$ *meets the same linear inverse constraint with* $f_1(j,i)$:

$$H \cdot f_2^{RND}(i,j) = HH^\dagger H f_2(i,j) + (H - HH^\dagger H)\hat{f}_2(i,j) = H \cdot f_2(i,j) = f_1(j,i). \tag{8}$$

Hence we replace $f_2$ in Eq. 5 with $f_2^{\text{RND}}$ in optimizing the node-wise interaction loss $\mathcal{L}^{\text{Inter}}$. The Thm. 1 give a formal guarantee of $f_2^{\text{RND}}$ reducing estimation error of $f_2$ for the **realness** constraint:

**Theorem 1** *Given any two nodes $i$ and $j$ with the ideal optimal learned function $f_2^*$, the error compared with $f_2^*(i,j)$ is defined as $L_2$-distance from it, where error for $f_2^{RND}$ and $f_2$ are correspondingly $e_0 = ||f_2^{RND}(i,j) - f_2^*(i,j)||^2$ and $e_1 = ||f_2(i,j) - f_2^*(i,j)||^2$. It holds that:*

$$||f_2^{RND}(i,j) - f_2^*(i,j)||^2 = ||(I - H^\dagger H)[f_2(i,j) - f_2^*(i,j)]||^2$$

*thus, $f_2^{RND}$ is superior than $f_2$ cause $e_0 \leq e_1$. (see appendix D for proof).*

**Extension to $n$-depth layer GNN.** Additionally, we scale the $f_1$ term in Eq. 3 up by a factor $\gamma$ into Eq. 9. $\gamma$ is derived from the summation of a geometric progression $\{1, \frac{1}{n_g}, \frac{1}{n_g^2}, \cdots, \frac{1}{n_g^{n-1}}\}$, where $n_g$ is the average value of all the unlearned nodes degrees. We would give an example of the typical two-layer GNN for illustration. Just before the first layer aggregation phase, their messages spread to $n_g$ 1-hop neighbors and approximately $n_g^2$ 2-hop neighbors before the second. Considering the share of influence passing of the second layer as 1, that of the first is $\frac{1}{n_g}$ for consistency, where $\gamma$ is the summation of the two. By analogy, the case of $n$ could be derived.

$$\tilde{h}_{e_i}^k = h_{e_i}^k - \gamma \sum_{j \in \mathcal{U} \cap \mathcal{N}_{e_i}} f_1(j,i) \tag{9}$$

**Local search loss.** Since the unlearning request typically involves a minor proportion of nodes, we extend $L^{\text{Inter}}$ into $L^{\text{Inter+}}$ to make it work as appendix C.1. The distribution of embeddings after unlearning is assumed to be near that before unlearning in minor removal. Therefore, for the retained node $p$ of high degrees, we propose to employ $\mathcal{L}^{\text{Local}}$ to search for the embedding after unlearning in a local area. Finally, the $\mathcal{L}^{\text{Local+}}$ is proposed for distillation in a general case as appendix C.2.

$$\mathcal{L}^{\text{Local}} = \sum_p KL(\text{Norm}[h_{e_p}^k], \text{Norm}[\tilde{h}_{e_p}^k]). \tag{10}$$

**Unlearning in a task.** To remove task-specific factors of targeted model, a simple gradient ascent loss $\mathcal{L}^+$ is adopted. For example, $\mathcal{L}^+$ could be maximizing classification loss upon unlearned nodes.

$$\mathcal{L}^+ = -\sum_{j \in \mathcal{U}} CE(\text{softmax}(\tilde{h}_{e_j}^k), y_{e_j}) \tag{11}$$

To sum up, the final loss at the given fixed unlearning ratio $\beta$ is:

$$\mathcal{L} = \beta \cdot (\mathcal{L}^+ + \mathcal{L}^{\text{Inter}}) + (1 - \beta) \cdot \mathcal{L}^{\text{Local}} \tag{12}$$

The intuition for the weight factor to be associated with the unlearning ratio is such that given a larger proportion of unlearning nodes, the local search term is less critical as the distribution gradually drifts away from the original one with more unlearned nodes. Meanwhile, the learning of node-wise interaction is more important and thus a higher weight is assigned.

Once trained, the new embedding for retained node is obtained by Eq. 9, which is the embedding when $\mathcal{U}$ is removed. Although we does not obtain a new GNN, it can be equivalently obtained by the original GNN rectified by $MLP_1$. It should also be noted that our method's privacy is robust as it does not involve any retraining and thus no original node attribute participates in the unlearning.

Table 1: Statistics of the datasets.

| Dataset | #Type | #Nodes | #Edges | #Features | #Classes |
|---------|-------|--------|--------|-----------|----------|
| Cora | Citation | 2,708 | 5,429 | 1,433 | 7 |
| Citeseer | Citation | 3,327 | 4,732 | 3,703 | 6 |
| CS | Coauthor | 18,333 | 163,788 | 6,805 | 15 |

## 5 EXPERIMENTS

We conduct experiments on three public graph datasets with different sizes, including Cora[1] (Kipf and Welling, 2017), Citeseer[2] (Kipf and Welling, 2017), and CS [3] (Zhang et al., 2021) in Table 1. These datasets are the benchmark datasets for evaluating the performance of GNN models for node classification task. Cora and Citeseer are citation datasets, where nodes represent the publications and edges indicate citations between two publications. CS is a coauthor dataset, where nodes are authors connected by an edge if they have collaborations; the features are keywords of the paper. For each dataset, we randomly split it into two subgraphs following the setting of GIF (Wu et al., 2023a): a training subgraph that consists of 90% nodes and the rest being the test subgraph.

**Implementation Detail.** Our experiments are implemented by `Pytorch` and conducted on a workstation with an NVIDIA GeForce RTX 4090 GPUs and 24GB memory. For node unlearning tasks, we randomly delete the nodes in the training graph with an unlearning ratio $\beta$, together with the connected edges. For graph model GCN, GAT, SGC, GIN, their training hyperparameters are: learning rate $\{0.05, 0.01, 0.05, 0.01\}$ and weight decay $\{1e^{-4}, 1e^{-3}, 1e^{-4}, 1e^{-4}\}$. The $H$ matrices for GCN, GAT, and SGC are set by the weight matrices in the convolutional and linear layers. GIN is particular in employing two linear layers at both the beginning and end of its convolutional layer. Thus, $H$ is set as the weight of the end one. The $H^{\dagger}$ is obtained by numpy packages.

**Metrics.** We comprehensively evaluate the performance of our method in terms of the following criteria proposed in Wu et al. (2023a) and Chen et al. (2022b):

• **Unlearning Efficiency**: We record the running time (RT) to reflect the unlearning efficiency across different unlearning algorithms for efficiency comparison.

• **Model Utility**: We use F1-score — the harmonic average of precision and recall — to measure the utility of unlearned models. This performance should be consistent with training from scratch, indicating an equivalent model utility after unlearning.

• **Unlearning Efficacy**: We adopt an indirect evaluation strategy that measures model utility in the task of removing adversarial or poisoned data.

• **Unlearning Privacy**: We employ the membership inference attack designed for unlearning to evaluate the privacy leakage of the unlearned samples by AUC values. Ideally, the attacker infers the unlearned samples with a probability close to random guessing.

**Baselines.** We compare our method with the following approaches: Retraining, the most straightforward solution that retrains the GNN model from scratch using only the remaining data. It achieves good model utility but falls short in unlearning efficiency. GIF: an extended version of the influence function on graph models, which considers the impact of the unlearned nodes. MEGU: the SOTA method as the learning-based baseline. More baselines and comparisons are also included in Tab. 4, 5, 6, and 7 of the appendix B.1 and appendix B.2.

---

[1]https://paperswithcode.com/dataset/cora

[2]https://paperswithcode.com/dataset/citeseer

[3]https://pytorch-geometric.readthedocs.io/en/latest/generated/torch_geometric.datasets.Coauthor.html

## 5.1 Unlearning Efficiency and Model Utility

It is noted that both the GIF method and our method are far more efficient than Retraining as shown in Table 2. Obviously, retraining consumes exorbitant time and computational expenses. With the increase in graph scale, retraining method suffers from $O(n^2)$ complexity with $n$ being the number of nodes. Thus, the retraining cost rises significantly on large graphs.

The efficiency of both our method and GIF is comparable on Cora and Citeseer. On larger graphs, GIF requires more iterations for convergence due to a more complicated Hessian matrix, thus leading to the performance gap in CS. In contrast, our method merely includes a low-cost training in embedding reconstruction and a 'subtraction' in the inference, and hence the computational cost increases almost linearly with the number of nodes. The efficiency of MEGU is between our method and GIF.

Table 2: Comparison of F1 scores and running time (RT) for different graph unlearning methods with 10% nodes deleted from the original graph. The bold indicates the best result for each GNN model on each dataset. Efficiency is discussed further in appendix C.4.

| Model | | Dataset | | | | | |
|---|---|---|---|---|---|---|---|
| | | Cora | | Citeseer | | CS | |
| Backbone | Strategy | F1-score (%) | RT (second) | F1-score (%) | RT (second) | F1-score (%) | RT (second) |
| GCN | Retrain | 80.81±1.11 | 6.33 | 69.73±1.37 | 8.53 | 90.39±0.32 | 108.25 |
| | GIF | 74.51±0.67 | 0.20 | 58.76±0.85 | 0.43 | 82.89±0.43 | 11.64 |
| | MEGU | 77.32±1.63 | 0.19 | 63.67±1.28 | **0.22** | 86.08±0.47 | 9.19 |
| | Ours | **82.73±1.00** | **0.16** | **67.75±0.90** | 0.47 | **90.91±0.40** | **1.23** |
| GAT | Retrain | 87.45±0.96 | 15.72 | 77.05±0.49 | 35.47 | 91.24±0.21 | 128.37 |
| | GIF | 81.74±0.75 | 1.39 | 75.58±0.46 | 1.49 | 89.21±0.01 | 6.74 |
| | MEGU | 85.81±1.03 | 0.94 | 79.05±0.79 | 1.23 | 88.43±0.58 | 4.51 |
| | Ours | **86.86±1.78** | **0.76** | **78.13±0.74** | **0.94** | **91.86±0.65** | **1.23** |
| SGC | Retrain | 80.29±1.80 | 6.63 | 69.49±1.49 | 8.29 | 89.29±1.77 | 110.48 |
| | GIF | 74.56±0.71 | 0.20 | 58.67±0.88 | **0.22** | 83.34±0.35 | 12.04 |
| | Ours | **81.84±0.43** | **0.14** | **65.89±0.56** | 0.40 | **90.26±0.49** | **0.96** |
| GIN | Retrain | 81.84±1.02 | 8.48 | 73.75±1.65 | 25.97 | 88.71±0.31 | 115.34 |
| | GIF | 75.17±1.99 | 0.79 | 68.49±1.73 | 1.67 | 87.68±1.19 | 2.33 |
| | Ours | **82.29±0.73** | **0.30** | **74.65±0.56** | **0.53** | **89.62±0.82** | **2.02** |

Compared with retraining, our method achieves almost an equivalent F1 score. The GIF method is less effective since the local structure surrounding the unlearned nodes is entirely changed. Both our method and GIF are well adapted to multiple backbones of GNNs. The MEGU's F1 scores of GCN and GAT are also included, which are better than GIF in the mass, but still inferior to our method.

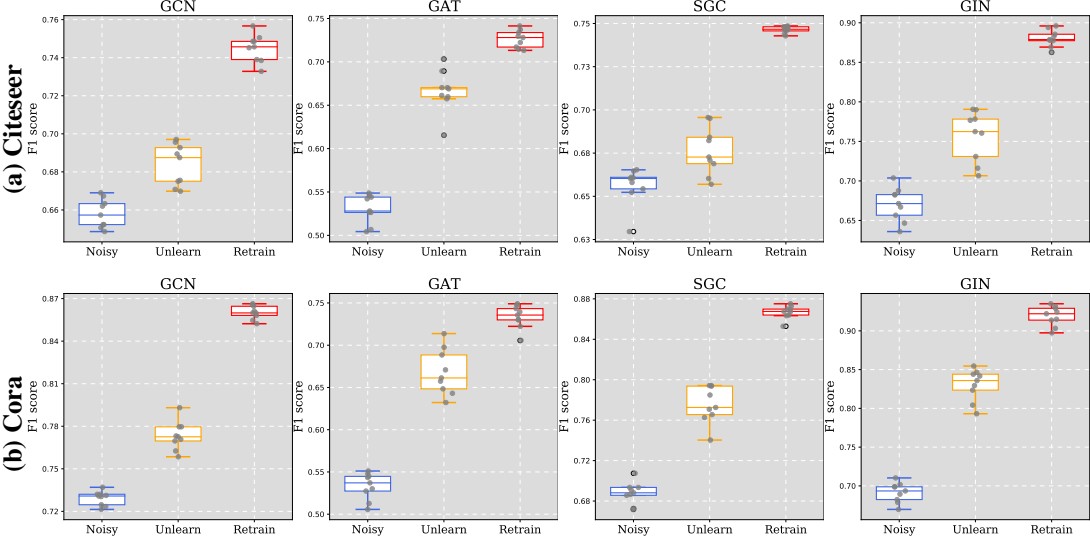

Figure 4: Unlearning efficacy box diagram. Grey data points are the statistics of 10 repeated results.

## 5.2 UNLEARNING EFFICACY

We test the unlearning efficacy by removing adversarial or poisoned nodes from the graph and evaluating the performance improvement. To create adversarial data, we randomly shift the labels of 30% training nodes to their groundtruth label plus 1, e.g. $y_{e_i} \rightarrow (y_{e_i} + 1) \mod n_c$, where $n_c$ denotes the number of categories. As shown in Fig. 9, we find that our proposed method can effectively improve the performance of the original model output representation on downstream tasks for different graph models. In the figure, models denoted as 'Noisy' are trained on all data, including the adversarial data, while those in 'Retrain' are trained from scratch with the adversarial data removed. Models in the 'Unlearn' category apply our method to remove the adversarial data.

## 5.3 UNLEARNING PRIVACY

Previous works suggested that unlearning would introduce additional privacy budget (Chen et al., 2021) since attackers could use the difference of embeddings before and after unlearning to infer about the unlearned nodes. Specifically, if a node is observed to have an abnormal gap in its embeddings before and after unlearning, it may be detected by the membership inference attack.

Table 3: Attack AUC of membership inference against our method ($\mathcal{A}_\text{I}$) and retraining ($\mathcal{A}_\text{II}$).

| Models | Cora | | Citeseer | | CS | |
|---|---|---|---|---|---|---|
| | $\mathcal{A}_\text{I}$ | $\mathcal{A}_\text{II}$ | $\mathcal{A}_\text{I}$ | $\mathcal{A}_\text{II}$ | $\mathcal{A}_\text{I}$ | $\mathcal{A}_\text{II}$ |
| GCN | 0.5014 | 0.5088 | 0.5010 | 0.5202 | 0.4988 | 0.4999 |
| GAT | 0.4857 | 0.5209 | 0.4916 | 0.5738 | 0.5015 | 0.5081 |
| SGC | 0.4825 | 0.5045 | 0.4963 | 0.5110 | 0.4994 | 0.5026 |
| GIN | 0.5129 | 0.5028 | 0.4987 | 0.5315 | 0.5002 | 0.5050 |

Note that our method is an embedding modification on the originals, so that the distribution of the difference may tell the unlearned samples off. Our membership inference attack follows the setting of Chen et al. (2021), where embeddings before and after unlearning 10% of all nodes are collected to calculate their $L_2$-norm distances. We try to tell the unlearned samples apart by setting varied distance thresholds, based on which we obtain the attack AUCs. Representative baselines comparisons are in the appendix B.3.

As shown in Table 9, the proposed unlearning method is resilient to membership inference attacks. Under various backbones and datasets, the AUC values are close to 0.5 consistently. It shows that the attack success rate is close to a random guess given the embeddings before and after unlearning. Hence, the unlearned nodes are indistinguishable from the retained nodes. In fact, even in the retraining method, AUCs deviate from 0.5 by some small margins, showing that even the ideal unlearning would leak the unlearned sample to some extent.

## 5.4 ABLATION STUDIES

**Range-null space decomposition (RND).** We show the effect of RND by removing the component from the framework at an unlearning ratio of 30%. The results on Cora and Citeseer illustrate that our method with RND achieves a closer F1 score with retraining for the multi-label node classification task. The F1 scores reduce when RND is removed (meaning that $f_2$ is used instead of $f_2^{\text{RND}}$). This shows that the linear inverse constraints lead to less loss of useful information in the unlearning process.

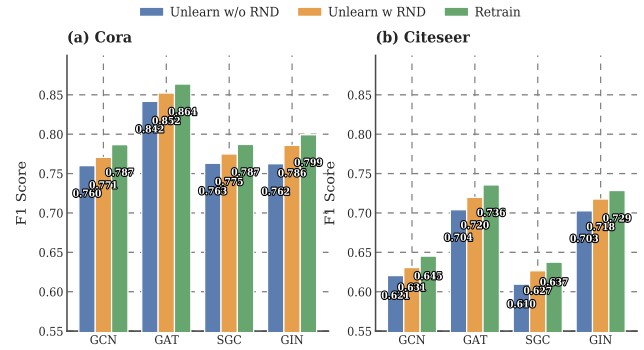

Figure 5: Ablation study of RND on Cora and Citeseer.

**Varying unlearning ratios.** We conduct experiments on Cora and Citeseer under unlearning ratios of 10%, 20%, 30%, and 40%, with results provided in Fig. 6. It is noted that the model utility of retraining decreases slowly and evenly with the increase of the unlearned proportion. The F1 score of GIF drops rapidly with the growing unlearning ratio. The trend of our method is consistent with that of retraining method, both in mean and variance, and thus it can be concluded that our method is more robust than GIF at different unlearning ratios.

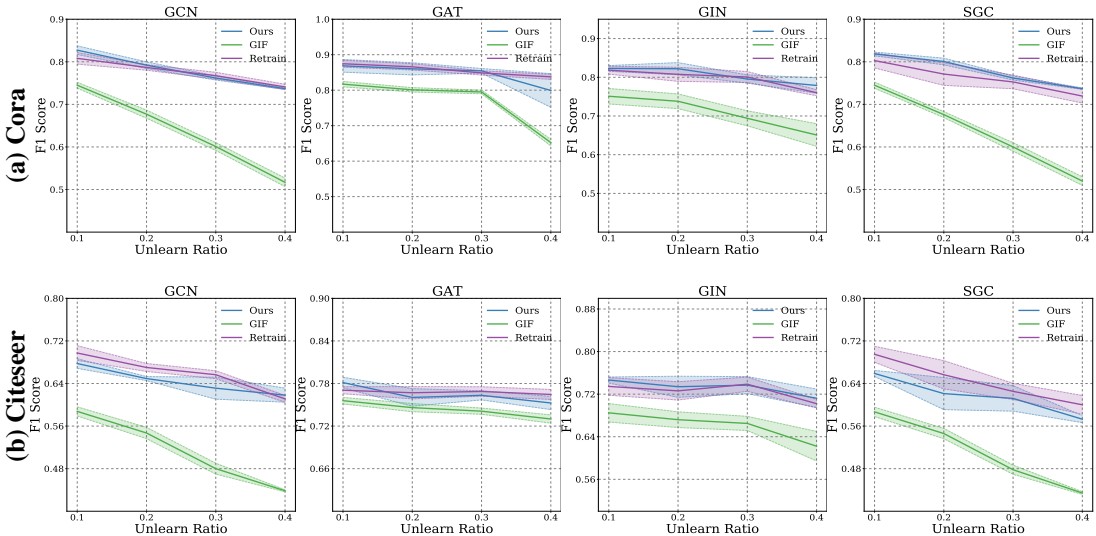

Figure 6: Varying unlearn ratios comparison with GIF and Retrain. Dotted line indicates variances.

## 5.5 UNLEARNING VISUALIZATION

For a more intuitive illustration, we employ 2D kernel density estimation to calculate the probability density function (PDF) $f_V$ for embedding set $V = (Mag_i, Ang_i)_{i=1}^n$. We use Gaussian kernel for PDF estimation, expressed as Eq. 13. In each domain(training from all, retraining, and our unlearning method), we independently derive the PDFs with bandwidth $h = 1$ as illustrated in Fig. 7. We verify that the embedding distribution of retraining overlaps with that training from all but significantly differs in density as shown in Fig. 7(a). Our method, demonstrated in Fig. 7(b), can produce the embedding distributions consistent with that of retraining.

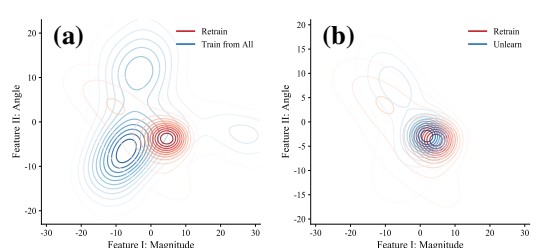

Figure 7: Embedding distribution under SGC when unlearning 30% of all nodes.

$$f_V(Mag, Ang) = \frac{1}{nh^2} \sum_{i=1}^{n} \frac{1}{2\pi} \exp \left\{ -\frac{1}{2h^2}[(Mag - Mag_i)^2 - (Ang - Ang_i)^2] \right\} \qquad (13)$$

## 6 CONCLUSION

We propose an efficient yet effective graph unlearning method in this paper. Our core idea is to reverse the aggregation process in GNN training by modeling the interaction between the unlearned nodes and their neighbors. Such an interaction is learned through $(k-1)$-th layer embedding reconstruction. The range-null space decomposition method is adopted to rectify the raw estimation of the interaction. Experimental results on multiple representative datasets demonstrate the effectiveness and efficiency of our proposed approach, compared to the state-of-the-art. This work also has important implications for interpretable graph unlearning.

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

# Appendix

## A  DETAILED RELATED WORK

### A.1  MACHINE UNLEARNING APPENDIX

Machine unlearning aims to eliminate the influence of a subset of the training data from the trained model out of privacy protection and model security, which could also remove the influence of noisy data on model performance. Ever since Cao and Yang (2015) first introduced the concept, several methods have been proposed to address the unlearning tasks, which can be classified into two branches: exact unlearning (Ginart et al., 2019; Karasuyama and Takeuchi, 2010; Bourtoule et al., 2021) and approximate unlearning (Koh and Liang, 2017; Guo et al., 2020; Izzo et al., 2021).

The former is aimed at creating models that perform identically to the model trained without the deleted data, or in other words, retraining from scratch, which is the most straightforward way but is computationally demanding. The *SISA* (Sharded, Isolated, Sliced, and Aggregated) approach (Bourtoule et al., 2021) partitions the data and separately trains a set of constituent models, which are afterward aggregated to form a whole model. During the procedure of unlearning, only the affected submodel is retrained with smaller fragments of data, thus greatly enhancing the unlearning efficiency.

The latter is designed for more efficient unlearning without retraining through fine-tuning the existing model parameters. Adapting the influence function (Koh and Liang, 2017) in the unlearning tasks, Guo et al. (2020) proposed to unlearn by removing the influence of the deleted data on the model parameters. Specifically, they used the deleted data to update models by performing a Newton step to approximate the influence of the deleted data and remove it, then they introduced random noise to the training objective function to ensure the certifiability. Unrolling SGD (Thudi et al., 2022) proposes a regularizer to reduce the 'verification error', which is an approximation to the distance between the unlearned model and a retrained-from-scratch model. The goal is to make unlearning easier in the future. Langevin Unlearning (Chien et al., 2024) leverages the Langevin dynamic analysis for the machine unlearning problem.

### A.2  GRAPH UNLEARNING APPENDIX

To provide a comprehensive understanding of Graph Unlearning, we categorize existing methodologies into four types based on their operational frameworks: (1) The Shards-based method: GraphEraser (Chen et al., 2022b) and GUIDE (Wang et al., 2023) extend the shards-based idea to graph-structured data, which offers partition methods to preserve the structural information and also designs a weighted aggregation for inference. Moreover, GraphRevoker (Zhang, 2024) utilized a property-aware sharding method and contrastive sub-model aggregation for efficient partial retraining and inference. They rely too much on a reasonable division of graph data into disjoint shards and these sub-models' costly retraining. Its inference cost is also higher as it requires results aggregation from all sub-models. (2) The IF-based method: like CGU (Chien et al.), GIF (Wu et al., 2023a), CEU (Wu et al., 2023b), IDEA (Dong et al., 2024), and GST (Pan et al., 2023) extends the influence-function method and proposes a similar formula for edge and node unlearning tasks on the graph model and further analyzes the theoretical error bound of the estimated influences under the Lipschitz continuous condition and loss convergence condition. These method work under the unlearning requests that minorly change the graph structure, like 10% of graph edges. Its performance drops for node unlearning. (3) Learning-based method: GNNDelete (Cheng et al., 2023) bounding edge prediction through a deletion operator and pays little attention to node embeddings equivalence. MEGU (Li et al., 2024) achieved effective and general graph unlearning through a mutual evolution design with adaptive HIN set selection. (4) Others: Projector (Cong and Mahdavi, 2023) provides closed-form solutions with theoretical guarantees but is more specialized in design. It couldn't be generalized easily.

Table 4: F1-score ± STD comparison under the standard setting of transductive node classification task with node unlearning request. The highest results are highlighted in bold while the second-highest results are marked with grey. Closer to Retrain's F1-score means better performance, and OOT indicates 'Out Of Time'.

| Strategy | Cora | | Citeseer | | CS | |
|---|---|---|---|---|---|---|
| | F1-score | RT (second) | F1-score | RT (second) | F1-score | RT (second) |
| Retrain | 0.8357±0.0161 | 5.96 | 0.6756±0.0132 | 7.69 | 0.8942±0.0147 | 102.46 |
| GraphEraser | 0.8114±0.0100 | 105.51 | 0.7357±0.0125 | 135.65 | 0.9124±0.0008 | 1323.26 |
| GUIDE | 0.7389±0.0218 | 44.80 | 0.6350±0.0071 | 57.60 | 0.8696±0.0015 | 2128.36 |
| GraphRevoker | 0.8109±0.0109 | 30.54 | 0.7345±0.0061 | 39.27 | 0.9126±0.0010 | 793.96 |
| GIF | 0.8175±0.0109 | 0.15 | 0.6258±0.0067 | 0.18 | 0.9187±0.0022 | 11.12 |
| CGU | 0.8637±0.0078 | 82.85 | 0.7562±0.0041 | 106.53 | OOT | OOT |
| ScaleGUN | 0.7882±0.0014 | 1.09 | 0.7342±0.0013 | 1.31 | 0.9144±0.0009 | 55.60 |
| IDEA | 0.8771±0.0025 | 0.14 | 0.6366±0.0049 | **0.17** | **0.8947±0.0022** | 33.35 |
| GNNDelete | 0.7478±0.0549 | 0.95 | 0.6426±0.0382 | 1.14 | 0.7626±0.0273 | 122.32 |
| MEGU | **0.8268±0.0156** | 0.95 | 0.6360±0.0111 | 1.15 | 0.9169±0.0008 | 123.35 |
| Projector | 0.8679±0.0237 | 5.86 | 0.7700±0.0065 | 7.03 | 0.8840±0.0063 | 88.97 |
| Ours | 0.8518±0.0143 | **0.14** | **0.6645±0.0068** | 0.40 | 0.9013±0.0089 | **0.96** |

## B    COMPREHENSIVE COMPARISON WITH EXISTING METHOD

Previous studies on GU have often employed varying dataset splits, different GNN backbones, and inconsistent unlearning request configurations, hindering direct comparisons between different methods. According to the existing benchmark OpenGU (Fan et al., 2025), we use datasets split into 80% for training and 20% for testing. For unlearning requests, 10% of nodes are selected for removal.

### B.1    OVERVIEW PERFORMANCES COMPARISON OF BASELINES

Regarding the overview comparison, we leverage SGC as a representative backbone of decoupled GNNs for the node unlearning task. We report the mean performance and standard deviation over 10 runs, ensuring consistency and reliability in the evaluation.

We compare our method with four types existing methods as Tab. 4. *The Shards-based method*: the additional subgraph partition cost is introduced, and the unlearned node subgraph retraining process also constitutes an important part of its time cost. Compared with training from scratch, their efficiency is very low and lacks model utility. This kind of method often sacrifices a part of graph structure in the process of community division, which leads to the forgetting of some useful information and the degradation of its performance.

*The IF-based method*: This types of method achieves comparable performance with retraining, this kind of method is the most practical one according to experimental verification. IDEA achieves the best performance in CS dataset, while lack of efficiency compared with ours. On the small-scale graph, its efficiency is competitive. However, due to the iterative approximation of hessian matrix of parameters, this kind of method will increase in time complexity at $O(n^2)$ with the expansion of graph nodes number. Although GIF has a gap with IDEA in performance, it has good scalability for it could be extended under multiple backbones and multiple task settings (Fan et al., 2025).

*The Learning-based method*: GNNdelete is more applicable in linear GNN models while MEGU achieves the best performance in Cora dataset. The latter introduced the concept of High-influence nodes for optimization, thus complicated interactions in large scale graph hinders the selection and optimization of such nodes. MEGU method is not good in model efficiency, compared with IF-based method.

*Others*: Projector is more specialized, limiting its scalability and generalization (Fan et al., 2025). The comparison of experimental data also shows that its performance and efficiency are not dominant compared with other types methods.

Our method has achieved good performance in multiple datasets and experimental settings. Compared with retraining, the utility of the our model has not been greatly affected. It has a very obvious

advantage in efficiency, especially on large-scale graphs CS. This fully demonstrates the practicability of this method.

## B.2    REPRESENTATIVE BASELINES PERFORMANCES COMPARISON

Representative methods that performed well in Tab. 4 are selected for comprehensive comparison. The same experiments have been conducted on the SGC backbone and the following Tab. 5 to Tab. 7 illustrate comparison on backbones of GAT, GCN, and GIN.

Our method shows superior performance on all three backbones and datasets. Compared with retrain, its performance has stable equivalence. It achieved the optimal or suboptimal results in all metrics. The highest results in the tables are highlighted in bold while the second-highest results are marked with grey. Closer to Retrain's F1-score means better performance.

Table 5: F1-score $\pm$ STD comparison with node unlearning request on GAT under standard setting.

| GAT | Cora | | Citeseer | | CS | |
|---|---|---|---|---|---|---|
| | F1-score | RT (second) | F1-score | RT (second) | F1-score | RT (second) |
| Retrain | 0.8745±0.0112 | 8.13 | 0.7854±0.0108 | 11.73 | 0.9425±0.0147 | 152.47 |
| MEGU | 0.8690±0.0143 | 0.48 | 0.7624±0.0145 | 0.51 | 0.9121±0.0132 | 64.32 |
| SGU | 0.8782±0.0117 | 0.43 | 0.7462±0.0126 | 0.41 | 0.9329±0.0100 | 41.30 |
| GNNDelete | 0.8480±0.0229 | 1.20 | 0.7310±0.0339 | 1.28 | 0.5554±0.0418 | 13.75 |
| GraphEraser | 0.8395±0.0101 | 11.60 | 0.7087±0.0103 | 11.12 | 0.9148±0.0141 | 37.69 |
| IDEA | 0.8413±0.0150 | 0.76 | 0.7312±0.0014 | 0.75 | 0.9177±0.0035 | 28.43 |
| Ours | 0.8695±0.0134 | 0.41 | 0.7895±0.0121 | 0.52 | 0.9400±0.0109 | 1.41 |

Table 6: F1-score $\pm$ STD comparison with node unlearning request on GCN under standard setting.

| GCN | Cora | | Citeseer | | CS | |
|---|---|---|---|---|---|---|
| | F1-score | RT (second) | F1-score | RT (second) | F1-score | RT (second) |
| Retrain | 0.8394±0.0104 | 5.234 | 0.7655±0.0109 | 7.654 | 0.9205±0.0148 | 99.64 |
| MEGU | 0.8727±0.0127 | 5.959 | 0.7042±0.0132 | 8.213 | 0.9046±0.0130 | 109.43 |
| SGU | 0.8875±0.0150 | 0.47 | 0.7450±0.0145 | 0.31 | 0.9190±0.0121 | 21.49 |
| GNNDelete | 0.8251±0.0316 | 0.86 | 0.7102±0.0218 | 0.91 | 0.4184±0.0351 | 36.73 |
| GraphEraser | 0.8376±0.0138 | 6.62 | 0.7486±0.0124 | 6.32 | 0.9010±0.0113 | 128.84 |
| IDEA | 0.8616±0.0011 | 0.35 | 0.6982±0.0041 | 0.33 | 0.9237±0.0042 | 32.43 |
| Ours | 0.8402±0.0143 | 0.32 | 0.7483±0.0100 | 0.28 | 0.9198±0.0106 | 1.13 |

Table 7: F1-score $\pm$ STD comparison with node unlearning request on GIN under standard setting.

| GIN | Cora | | Citeseer | | CS | |
|---|---|---|---|---|---|---|
| | F1-score | RT (second) | F1-score | RT (second) | F1-score | RT (second) |
| Retrain | 0.8358±0.0134 | 9.25 | 0.7502±0.0144 | 7.52 | 0.8998±0.0122 | 130.84 |
| MEGU | 0.8007±0.0107 | 0.21 | 0.7072±0.0112 | 0.33 | 0.8416±0.0139 | 72.15 |
| SGU | 0.8690±0.0149 | 0.31 | 0.7492±0.0137 | 0.32 | 0.8939±0.0102 | 18.47 |
| GNNDelete | 0.7343±0.0320 | 0.82 | 0.6922±0.0405 | 0.77 | 0.3627±0.0471 | 2.20 |
| GraphEraser | 0.8542±0.0111 | 7.17 | 0.7568±0.0129 | 5.77 | 0.0425±0.0126 | 21.03 |
| IDEA | 0.8137±0.0046 | 0.42 | 0.6997±0.0033 | 3.40 | 0.8391±0.0110 | 32.04 |
| Ours | 0.8429±0.0125 | 0.28 | 0.7652±0.0115 | 0.22 | 0.9022±0.0136 | 1.03 |

## B.3    REPRESENTATIVE BASELINES PRIVACY COMPARISON

In the main body, we only provide the equivalence of the proposed method with the retraining by showing the closeness of their AUC values. We have conducted additional comparison experiments in Tab. 8 to illustrate that our method has preserved more privacy than other methods. It achieved SOTA in Citeseer and CS on all four backbones, while the performances on Cora are nearly optimal with less than 0.02 gap.

Table 8: Attack AUC of membership inference against different unlearning methods.

| Models | Cora | | | | Citeseer | | | | CS | | | |
|---|---|---|---|---|---|---|---|---|---|---|---|---|
| | GCN | GAT | SGC | GIN | GCN | GAT | SGC | GIN | GCN | GAT | SGC | GIN |
| GIF | 0.5952 | 0.5428 | 0.5834 | 0.6010 | 0.6052 | 0.5381 | 0.5996 | 0.5657 | 0.5744 | 0.5977 | 0.5591 | 0.5930 |
| MEGU | 0.4856 | **0.5008** | 0.5081 | 0.4768 | 0.4532 | 0.4483 | 0.4735 | 0.4750 | 0.5155 | 0.5190 | 0.5186 | 0.4964 |
| SGU | 0.5019 | 0.5084 | **0.5015** | **0.5026** | 0.4637 | 0.4366 | 0.4856 | 0.5216 | 0.5048 | 0.5124 | 0.5208 | 0.4962 |
| GraphEraser | 0.7601 | 0.5358 | 0.7684 | 0.4694 | 0.7991 | 0.6574 | 0.7521 | 0.5950 | 0.4982 | 0.5192 | 0.6582 | 0.5736 |
| IDEA | 0.5371 | 0.4754 | 0.5328 | 0.5344 | 0.5672 | 0.5279 | 0.5659 | 0.5075 | 0.5021 | 0.4827 | **0.4970** | 0.5193 |
| Ours | **0.5014** | 0.4857 | 0.4825 | 0.5129 | **0.5010** | **0.4916** | **0.4963** | **0.4987** | **0.5010** | **0.4916** | 0.4963 | **0.4987** |

## C  METHOD EXTENSION

### C.1  LOW RATIO REMOVAL

As we all know, machine learning methods rely on a sufficient training data to take effect. As we model unlearning as a data-driven machine learning problem, to provide sufficient training data cannot be ignored. Specifically, sufficient unlearned nodes for embeddings reconstruction towards training of the embeddings modification model is important. When the number of nodes to be forgotten is very small, node interactions are hard to learn according to Eq. 5. Considering that our unlearning method training and inference are strongly decoupled, all nodes besides unlearned nodes, denoted as $\mathcal{R}$, also satisfy the interaction between nodes. The loss function of interaction learning of $\mathcal{L}^{\text{Inter}}$ could be extended as follows $\mathcal{L}^{\text{Inter+}}$:

$$\mathcal{L}^{\text{Inter+}} = \frac{1}{|m|} \sum_{m \in \mathcal{U} \cup \mathcal{R}} KL\left(\text{Norm}[\boldsymbol{h}_{e_m}^{k-1}], \text{Norm}[\sum_i f_2(i,m)]\right) \tag{14}$$

This $\mathcal{L}^{\text{Inter+}}$ loss function is attached to our experiment for implementation, and the results in the main body are obtained.

### C.2  HIGH RATIO REMOVAL

When the ratio of forgotten nodes is very large, like higher than 30%, the graph structural factors play a major role in embedding distribution, as our experimental experience.

In order to adapt to this scene, we modified the formula of $\tilde{\boldsymbol{h}}_{e_i}^k$ to achieve better forgetting. With the help of the training graph $G' = G/\Delta G$ after deleting the unlearned nodes, we get the middle embedding through direct inference $\overline{\boldsymbol{h}}_{e_i}^k = \text{GNN}_0(G')$, where $\text{GNN}_0(\cdot)$ is the model before unlearning. Thus, $\tilde{\boldsymbol{h}}_{e_i}^k$ is given by:

$$\tilde{\boldsymbol{h}}_{e_i}^k = \overline{\boldsymbol{h}}_{e_i}^k - \gamma \sum_{j \in \mathcal{U}} f_1(j,i) \tag{15}$$

By the way, the **Local search loss** is also transformed into:

$$\mathcal{L}^{\text{Local+}} = \sum_p KL(\text{Norm}[\overline{\boldsymbol{h}}_{e_i}^k], \text{Norm}[\tilde{\boldsymbol{h}}_{e_p}^k]). \tag{16}$$

At this time, if we still employ embedding $\boldsymbol{h}_{e_i}^k = \text{GNN}_0(G)$ and obtain $\tilde{\boldsymbol{h}}_{e_i}^k$ by Eq. 9, the produced embedding distribution is not equivalent to retraining. As shown in Fig. 8(a), the embedding distribution of retrain is far different from training from all. The PDFs of the two have clearly deviated in angle. While the obtained unlearned distribution almost overlaps with that of training from all in Fig. 8(b).

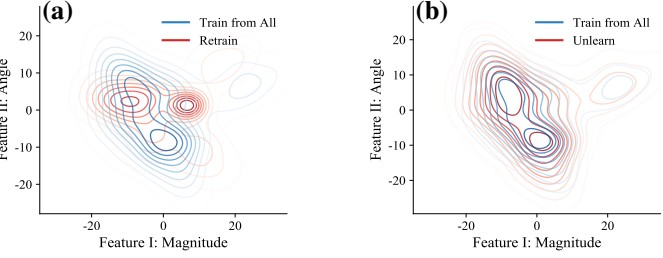

Figure 8: Embeddings distribution of SGC unlearning 40% nodes.

It is challenging to unlearn with a huge distribution gap before and after retraining through a small disturbance. Therefore, it is better to use a middle result with attached structural factors for an adaptation.

## C.3 EDGE UNLEARNING

we simply use an MLP layer named $MLP_1$ to represent the interaction to be removed in $\boldsymbol{h}_{e_i}^k$ in the $k$-th layer between remained node $e_i$ and unlearned node $e_j$. It should be noted that when the above $f_1$ is subtracted from $\boldsymbol{h}_{e_i}^k$, the influence of an edge $e_j \rightarrow e_i$ is actually eliminated.

Thus, given unlearning edge target as $\mathcal{U}_e$, edge unlearning may be extended in inference as:

$$\tilde{\boldsymbol{h}}_{e_i}^k = \boldsymbol{h}_{e_i}^k - \gamma \sum_{<j,i>\in\mathcal{U}_e} f_1(j,i) \tag{17}$$

## C.4 REPRESENTATIVE BASELINES EFFICIENCY COMPARISON

The memory consumption required by our proposed method in various tasks and corresponding FLOPs is illustrated below, which could be conducted with an NVIDIA GeForce RTX 4090 GPUs.

Table 9: Maximum Memory consumption and corresponding FLOPs of the proposed method.

| Models | Cora | | Citeseer | | CS | |
|---|---|---|---|---|---|---|
| | MEM | FLOPs | MEM | FLOPs | MEM | FLOPs |
| GCN | 0.73 GB | 62,938,717(+0.45%) | 1.02 GB | 197,933,004(+0.12%) | 4.76 GB | 2,010,758,706(+0.31%) |
| GAT | 0.78 GB | 502,307,745(+0.06%) | 1.09 GB | 1,582,470,428(+0.01%) | 15.94 GB | 16,050,937,830(+0.04%) |
| SGC | 0.73 GB | 62,852,061(+0.45%) | 1.02 GB | 197,826,540(+0.12%) | 5.00 GB | 2,010,172,050(+0.31%) |
| GIN | 0.69 GB | 65,845,867(+0.43%) | 0.90 GB | 201,491,836(+0.11%) | 4.12 GB | 2,030,200,248(+0.31%) |

We also conduct comparison experiments with representative methods concerning FLOPs. We listed the extra overhead of FLOPs caused by the unlearning process. FLOPs could be one depiction of a model's complexity. However, the actual running time of the model is affected by many factors.

Table 10: Comparisons on Cora of backbones FLOPs with extra FLOPs brought by unlearning.

| Models | Cora | | | |
|---|---|---|---|---|
| | GCN | GAT | SGC | GIN |
| Base FLOPs | 62,657,238 | 65,564,388 | 62,570,582 | 502,026,266 |
| MEGU | 101,018,358(+61.22%) | 540,387,386(+7.64%) | 100,931,702(+61.31%) | 103,925,458(+58.51%) |
| SGU | 71,868,338(+14.70%) | 543,569,766(+8.28%) | 69,176,682(+10.56%) | 74,841,088(+14.15%) |
| GNNdelete | 78,400,338(+25.13%) | 517,769,366(+3.14%) | 78,313,682(+25.16%) | 81,307,488(+24.01%) |
| GraphEraser | 413,486,913(+559.92%) | 852,855,941(+69.88%) | 413,400,257(+560.69%) | 416,394,063(+535.09%) |
| IDEA | 53,252,980,14(+8399.09%) | 3,412,835,523,079(+679712.14%) | 10,138,775,860(+16103.74%) | 54,779,580,131(+83450.81%) |
| Ours | **62,938,717(+0.45%)** | **502,307,745(+0.06%)** | **62,852,061(+0.45%)** | **65,845,867(+0.43%)** |

Table 11: Comparisons on Citeseer of backbones FLOPs with extra FLOPs brought by unlearning.

| Models | Citeseer | | | |
|---|---|---|---|---|
| | GCN | GAT | SGC | GIN |
| Base FLOPs | 197,703,982 | 201,262,814 | 197,597,518 | 1,582,241,406 |
| MEGU | 230,398,222(+16.54%) | 1,614,935,646(+2.07%) | 30,291,758(+16.55%) | 233,957,054(+16.24%) |
| SGU | 215,030,182(+8.76%) | 1,682,725,206(+6.35%) | 207,495,318(+5.01%) | 218,651,814(+8.64%) |
| GNNdelete | 212,200,982(+7.33%) | 1,596,738,406(+0.92%) | 212,094,518(+7.34%) | 215,759,814(+7.20%) |
| GraphEraser | 938,558,471(+374.73%) | 2,323,095,895(+46.82%) | 938,452,007(+374.93%) | 942,117,303(+368.10%) |
| IDEA | 352630,204289(+1.78*$10^5$) | 22,578716,996653(+1.42*$10^6$) | 49588,334115(+2.49*$10^4$) | 356371,809091(+1.76*$10^5$) |
| Ours | **197,933,004(+0.12%)** | **1,582,470,428(+0.01%)** | **197,826,540(+0.12%)** | **201,491,836(+0.11%)** |

FLOPs wouldn't absolutely rise and fall together with running time (RT). As one of the typical metrics to illustrate model complexity, it is also listed as Tab. 10 to Tab. 12, which shows that our method is efficient in floating-point operations.

## D PROOF FOR THEOREM 1

Proof for $f_2^{\mathrm{RND}}$ is superior to the preliminary one $f_2$:

Table 12: Comparisons on CS of backbones FLOPs with extra FLOPs brought by unlearning.

| Models | CS | | | |
|---|---|---|---|---|
| | GCN | GAT | SGC | GIN |
| Base FLOPs | 2,004,447,315 | 2,023,888,857 | 2,003,860,659 | 16,044,626,439 |
| MEGU | 3,222,523,515(+60.77%) | 17,262,702,639(+7.59%) | 3,221,936,859(+60.79%) | 3,241,965,057(+60.18%) |
| SGU | 2,066,236,915(+3.08%) | 16,259,263,639(+1.34%) | 2,064,238,059(+3.01%) | 2,085,780,857(+3.06%) |
| GNNdelete | 2,259,234,315(+12.71%) | 16,299,413,439(+1.59%) | 2,258,647,659(+12.71%) | 2,278,675,857(+12.59%) |
| GraphEraser | 31,968,836,229(+1494.90%) | 46,009,015,353(+186.76%) | 31,968,249,573(+1495.33%) | 31,988,277,771(+1480.54%) |
| IDEA | $1,193399,393354(+5.94*10^4)$ | $76,296883,845668(+4.75*10^5)$ | $1,044241,501293(+5.20*10^4)$ | $1,204622,114256(+5.94*10^4)$ |
| Ours | **2,010,758,706(+0.31%)** | **16,050,937,830(+0.04%)** | **2,010,172,050(+0.31%)** | **2,030,200,248(+0.31%)** |

Assuming $f_2^*$ is the optimal learned function, we prove that the $L_2$-distance of $f_2^{\text{RND}}$ and $f_2^*$ is smaller than that of $f_2$ and $f_2^*$. Firstly, it holds that $f_1 = H f_2^*$ according to our modeling and $(H^\dagger H)^\top = H^\dagger H$ holds for $H^\dagger$'s property as *generalized inverse matrix*.

With the above conditions, the $L_2$-distance between them could be bridged by:

$$||f_2^{\text{RND}} - f_2^*||^2 = ||H^\dagger f_1 + (I - H^\dagger H)f_2 - f_2^*||^2$$
$$= ||(H^\dagger H - I)f_2^* + (I - H^\dagger H)f_2||^2$$
$$= ||(I - H^\dagger H)(f_2 - f_2^*)||^2$$

Then, we prove that $A = (I - H^\dagger H)$ is *projection matrix* thus the $L_2$-norm of $(f_2 - f_2^*)$ is smaller after its projection. Equivalently, we show that by verifying $A$ as both *idempotent matrix* and *symmetric matrix*.

(1) ***Idempotent Matrix***. It holds that $AA = A$ thus $A$ is *idempotent matrix*:

$$AA = (I - H^\dagger H)(I - H^\dagger H)$$
$$= I - H^\dagger H - H^\dagger H + H^\dagger H H^\dagger H$$
$$= I - H^\dagger H - H^\dagger H + H^\dagger H$$
$$= I - H^\dagger H = A$$

(2) ***Symmetric Matrix***. It holds that $A = A^\top$ thus $A$ is *symmetric matrix*:

$$A^\top = (I - H^\dagger H)^\top$$
$$= I - (H^\dagger H)^\top$$
$$= I - H^\dagger H = A$$

Thus, $A$ is *projection matrix* and the proof is completed. The estimation error between $f_2^{\text{RND}}$ and $f_2^*$ is smaller than that of $f_2$ and $f_2^*$. $f_2^{\text{RND}}$ is superior to the preliminary one $f_2$.

### D.1 DISCUSSION OF DEEPER GNN UNLEARNING

When the GNN has deeper layers, the message of each node passes through the whole graph. embeddings of one node will inevitably aggregate the information in the whole graph and suffer from over-smoothing (Xu et al., 2018). Therefore, the number of layers will not be very deep, which is 2 or 3 under normal circumstances.

At the same time, according to the settings in Luo et al. (2024), the deeper GNN (10 layers at most) carried out dimension reduction only in the input and output layers, while the hidden layers' embedding dimension remains unchanged. The hidden layers' embeddings may not be that different.

Our method still works in design; however, we suggest modeling a deeper model into one GNN with 2-layer and 3-stage embeddings for better handling and explanation. The 3-stage embeddings are:

$$[\text{initial features}] \Rightarrow [\text{the middle hidden layer features}] \Rightarrow [\text{output features}]$$

The middle layer is far from that of the input and output layer, thus it's more distinguishable, promoting reconstruction effects.

As the 3-stage embeddings are determined, the stacked inter-layer between the adjacent stage embeddings could be treated as one composite layer, so as to get its $H$ value. The problem has been regulated and could be resolved similarly.

# E  ALGORITHM PSEUDOCODE

The algorithm removes the influence of unlearned nodes from GNN embeddings by Node Interaction Learning and Unlearning Inference in two phases. It involves two MLPs, $f_1$ and $f_2$, with three losses for optimization. All mathematical symbols are consistent with the main body.

---

**Algorithm 1** Graph Unlearning via Reconstruction Algorithm

---

**Require:** Trained GNN model $f_G$ with frozen parameters for initial embeddings generation, graph $G = (\mathcal{V}, \mathcal{E})$, unlearning node set $\mathcal{U}$, unlearning ratio $\beta$, $k$-th layer embeddings $\boldsymbol{h}^k$, $(k-1)$-th layer embeddings $\boldsymbol{h}^{k-1}$.

**Ensure:** Unlearned node embeddings $\tilde{\boldsymbol{h}}^k$

1: **Phase 1: Learning Node Interactions**
2: Initialize MLPs $f_1$ and $f_2$
3: Extract weight matrix $W^k$ from the $k$-th layer of GNN, set $H = W^k$
4: Compute generalized inverse matrix $H^\dagger$ of $H$
5: Compute average degree: $n_g = \frac{1}{|\mathcal{U}|} \sum_{j \in \mathcal{U}} |\mathcal{N}_{e_j}|$
6: Compute scaling factor: $\gamma = \sum_{t=0}^{n-1} \frac{1}{n_g^t}$, where $n$ is the number of GNN layers
7: **while** not converged **do**
8:    **for** each unlearned node $m \in \mathcal{U}$ **do**
9:      **for** each retained node $i \in \mathcal{N}_m$ **do**
10:        Compute node interaction: $f_1(m, i) = \text{MLP}_1(\boldsymbol{h}_{e_m}^{k-1}, \boldsymbol{h}_{e_i}^{k-1})$
11:        Preliminary estimation of reverse interaction: $\hat{f}_2(i, m) = \text{MLP}_2(f_1(m, i))$
12:        Range-Null Space Decomposition: $f_2^{\text{RND}}(i, m) = H^\dagger f_1(m, i) + (I - H^\dagger H)\hat{f}_2(i, m)$
13:      **end for**
14:      Aggregate neighbor information: $\hat{\boldsymbol{h}}_{e_m}^{k-1} = \sum_{i \in \mathcal{N}_{e_m}} f_2^{\text{RND}}(i, m)$
15:    **end for**
16:    Compute interaction reconstruction loss: $\mathcal{L}^{\text{Inter}} = \frac{1}{|\mathcal{U}|} \sum_{m \in \mathcal{U}} \text{KL}(\text{Norm}[\boldsymbol{h}_{e_m}^{k-1}], \text{Norm}[\hat{\boldsymbol{h}}_{e_m}^{k-1}])$
17:    Compute $\tilde{\boldsymbol{h}}_{e_i}^k = \boldsymbol{h}_{e_i}^k - \gamma \sum_{j \in \mathcal{U} \cap \mathcal{N}_{e_i}} f_1(j, i)$
18:    Compute local search loss: $\mathcal{L}^{\text{Local}} = \sum_p \text{KL}(\text{Norm}[\boldsymbol{h}_{e_p}^k], \text{Norm}[\tilde{\boldsymbol{h}}_{e_p}^k])$
19:    Compute task-specific loss: $\mathcal{L}^+ = -\sum_{j \in \mathcal{U}} \text{CE}(\text{softmax}(\tilde{\boldsymbol{h}}_{e_j}^k), y_{e_j})$
20:    Compute total loss: $\mathcal{L} = \beta \cdot (\mathcal{L}^+ + \mathcal{L}^{\text{Inter}}) + (1 - \beta) \cdot \mathcal{L}^{\text{Local}}$
21:    Backpropagate to update parameters of $f_1$ and $f_2$
22: **end while**
23: **Phase 2: Unlearning Inference**
24: **for** each retained node $i \in \mathcal{V} \setminus \mathcal{U}$ **do**
25:    Compute unlearned embedding:
26:    $\tilde{\boldsymbol{h}}_{e_i}^k = \boldsymbol{h}_{e_i}^k - \gamma \sum_{j \in \mathcal{U} \cap \mathcal{N}_{e_i}} f_1(j, i)$
27: **end for**
28: **return** $\tilde{\boldsymbol{h}}^k$

---

# F  THE USE OF LARGE LANGUAGE MODELS (LLMS)

Large Language Models were not used in the preparation of this manuscript. The LLMs did not contribute to research ideas, experimental design, and data analysis. All technical results, claims, and conclusions remain the sole responsibility of the authors.

# G  REPRESENTATIVE BASELINES EFFICACY COMPARISON

We compare the efficacy performances of GIF and MEGU methods with our proposed method. The experimental setup is the same as that in 5.2, and the results show that our method is significantly superior to the other two methods in removing adversarial or poisoned nodes, which is closer to that of retraining.

There are models denoted as 'Noisy' trained on all data, including the adversarial ones, as 5.2. These are not shown here for data clarity and conciseness. It is worth noting that only our method achieves an improvement over 'Noisy' in almost all backbones and datasets, which further reveals the problems existing in the current graph unlearning method.

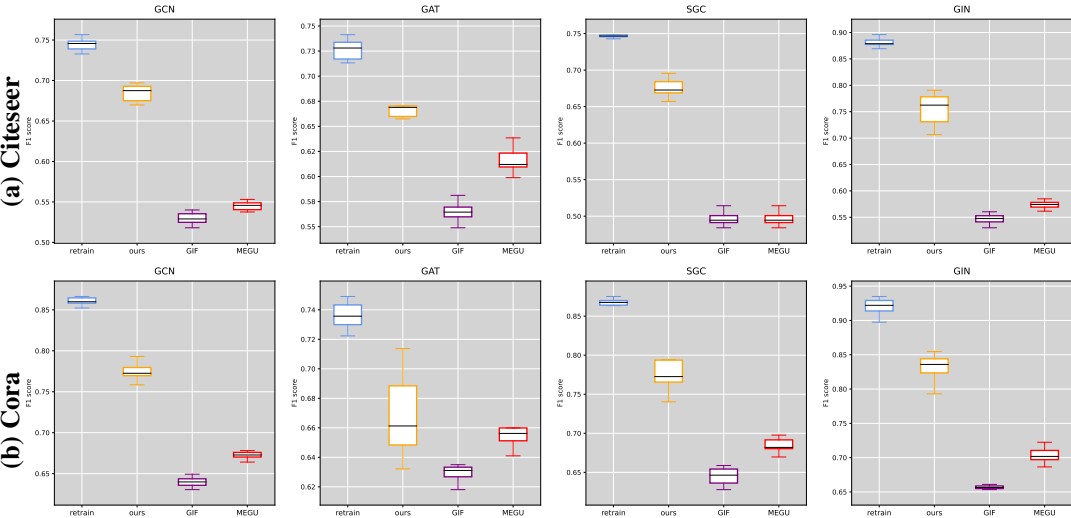

Figure 9: Unlearning efficacy comparison of the our method with GIF and MEGU in box diagram.

