# OpenReview forum: "Graph Unlearning via Reconstruction --- A Range-Null Space Decomposition Approach"
_ICLR.cc/2026/Conference — Submitted to ICLR 2026_

### Official Review · Reviewer_htoX · 2025-10-29

**Soundness:** 3
**Presentation:** 3
**Contribution:** 3
**Rating:** 6
**Confidence:** 3

**Summary:**

This paper proposes a novel graph unlearning method that reverses GNN aggregation by learning node-wise interactions through embedding reconstruction. Rather than retraining the entire model or modifying all parameters, the method learns two MLPs to model interactions between unlearned and retained nodes, applies range-null space decomposition to rectify estimation errors with theoretical guarantees (Theorem 1), and modifies only the output embeddings to eliminate the influence of unlearned nodes. Experiments on three datasets (Cora, Citeseer, CS) with four GNN backbones (GCN, GAT, SGC, GIN) demonstrate 40 to 88× speedup compared to retraining while achieving comparable F1 scores, strong unlearning efficacy on poisoned data, and privacy preservation under membership inference attacks.

**Strengths:**

1. **Novel approach with theoretical guarantees**. The core insight of reversing GNN message passing through reconstruction is elegant, and the range-null space decomposition (Equation 7, Theorem 1) provides formal guarantees that f_2^{RND} reduces L2-norm error compared to the raw f2 estimation. This addresses a fundamental challenge in graph unlearning where the entanglement of node features makes influence estimation difficult. The method's design naturally handles the under-determined reconstruction problem from lower to higher dimensional embeddings.

2. **Strong empirical performance with good generalizability**. The method achieves substantial efficiency gains (Table 2 shows 0.14-2.02 seconds vs 6-128 seconds for retraining on various models) while maintaining utility close to retraining across different unlearning ratios (Figure 6 shows consistent trends from 10% to 40%). The approach works uniformly across multiple GNN architectures without architecture-specific modifications, and demonstrates superior robustness compared to GIF whose performance degrades rapidly with increasing unlearning ratios (Figure 6 shows GIF dropping from 0.75 to 0.5 F1 score while the proposed method maintains 0.72-0.82).

3. **Effective visualization demonstrates embedding distribution equivalence**. The 2D kernel density estimation visualizations (Figures 7 and 8) provide intuitive evidence that the method produces embedding distributions matching retraining. Figure 7(b) shows the unlearned distribution closely overlaps with retraining at 30% ratio, while Figure 8(b) demonstrates the modified approach maintains this property at 40% ratio.

**Weaknesses:**

1. **Insufficient theoretical justification for key design choices**. While Theorem 1 proves f_2^{RND} has smaller error than f2, the paper lacks analysis of when and why reconstructing specifically the (k-1)-th layer embeddings is optimal. The justification on page 4 (lines 209) merely states it "balances the amount of information to be learned and that to be forgotten" without formal analysis.

2. **The method requires case-specific formulations that undermine generality**. The core approach fails at different unlearning scenarios without ad-hoc modifications: L^Inter extends to L^Inter+ for low ratios (Equation 14, Appendix C.1), embeddings switch from h^k to h̄^k with inference on modified graph G' for ratios above 30% (Equations 9 to 15, Appendix C.2), and edge unlearning requires separate handling (Equation 17, Appendix C.3).

**Questions:**

1. Could you provide theoretical or empirical evidence showing that (k-1)-th layer reconstruction is optimal compared to other layers?

2. Recent work has shown that unlearning methods can be vulnerable to sophisticated unlearning inversion attacks that exploit the differences between original and unlearned models to reconstruct unlearned data [1,2,3]. Since your method explicitly models node interactions and stores them in learned MLPs, is your approach robust to these inversion attacks? Can you evaluate attacks like those in Hu et al. [1] for general machine unlearning, Zhang et al. [2] specifically for GNN unlearning inversion, and Wu and Wang [3] for attacking incomplete unlearning?

3. Input encryption techniques [4,5,6] are widely used to protect sensitive data during model training. Could you briefly compare how graph unlearning differs from input encryption in terms of privacy guarantees and practical deployment?

### References

[1] Hongsheng Hu, Shuo Wang, Tian Dong, Minhui Xue. “Learn what you want to unlearn: Unlearning inversion attacks against machine unlearning”. IEEE S&P 2024.

[2] Jiahao Zhang, Yilong Wang, Zhiwei Zhang, Xiaorui Liu, Suhang Wang. “Unlearning Inversion Attacks for Graph Neural Networks”. WSDM 2026.

[3] Kun Wu, Wendy Hui Wang. “Verification of Incomplete Graph Unlearning through Adversarial Perturbations”. KDD 2025.

[4] Yangsibo Huang, Zhao Song, Kai Li, Sanjeev Arora. “InstaHide: Instance-hiding Schemes for Private Distributed Learning”. ICML 2020.

[5] Yangsibo Huang, Zhao Song, Danqi Chen, Kai Li, Sanjeev Arora. “TextHide: Tackling Data Privacy in Language Understanding Tasks”. EMNLP 2020.

[6] Sitan Chen, Xiaoxiao Li, Zhao Song, Danyang Zhuo. "On InstaHide, Phase Retrieval, and Sparse Matrix Factorization". ICLR 2021.

---

> ### Author Response · Authors · 2025-11-14
>
> ```
> W1. Insufficient theoretical justification for key design choices. While Theorem 1 proves f_2^{RND} has smaller error than f2, the paper lacks analysis of when and why reconstructing specifically the (k-1)-th layer embeddings is optimal.
> ```
>
> W1: Thanks for your feedback. We would like to give a more precise explanation.
> 1. Reconstructing former high-dimensional layers is a harder under-determined problem with exponentially growing solution space, making it computationally intractable and prone to error in implementation.
> 2. Former layer embeddings of $(k-1)$-th related to initial features, thus more risk in revealing information.
> 3. Reconstructing $k$-th embeddings leads to excessive unlearning. Because after aggregation of GNN, the target of reconstruction, unlearned node embeddings, maintains more useful information shouldn't be forgotten from retained nodes.
>
> ```
> W2. The method requires case-specific formulations that undermine generality.
> ```
>
> W2: Thanks for your thorough review, the details you pay attention to are undoubtedly very important. However, we want to clarify that our method is implemented through a unified framework. These extensions are principled adaptations to different scenarios, not ad-hoc modifications. They are proposed under specific problems, but apply to all situations.
>
> 1. $L^\text{Inter}$ is extended to $L^\text{Inter+}$ for a unified implementation of different unlearning ratios as stated in lines 780-781.
> 2. When a large proportion of nodes is removed, the graph structure changes significantly. Using the modified graph $G^{'}$ for inference is necessary cause the original graph structure no longer represents the unlearning scenario. This is not an ad-hoc modification but a principled adaptation to the changed graph topology. In practice, we only need to change the input of GNN in the implementation, which is also widely used in our experiments.
> 3. Edge unlearning is a different task from node unlearning, requiring different handling, while we focus on node unlearning in this paper. But our framework naturally extends to edges by modifying the interaction modeling to account for edge removal.
>
> ```
> Q1. Theoretical or empirical evidence showing that (k-1)-th layer reconstruction is optimal compared to other layers?
> ```
>
> Q1: Refer to W1.
>
> ```
> Q2. Unlearning methods can be vulnerable to sophisticated unlearning inversion attacks that exploit the differences between original and unlearned models to reconstruct unlearned data [1,2,3]. Since your method explicitly models node interactions and stores them in learned MLPs, is your approach robust to these inversion attacks? Can you evaluate attacks like those in Hu et al. [1] for general machine unlearning, Zhang et al. [2] specifically for GNN unlearning inversion, and Wu and Wang [3] for attacking incomplete unlearning?
> ```
>
> Q2: Thanks for your in-depth question. Their attack assumptions do not hold; thus, our approach is robust to these inversion attacks.
>
> [1] employs model parameter differences, but our GNN parameters are frozen during unlearning. The differences could only be obtained in the embedding, thus [1] could not take effect. [2, 3] focused on edge/link unlearning, while ours resolved node unlearning. [3] is declared to extend to node unlearning by injecting fake nodes and edges into the graph data, with no relevant experiments displayed. Meanwhile, its source code contains nothing related to node unlearning, which blocked our attempt to verify.
>
> At the same time, our method's privacy is evaluated through [a], which is actually cited in [1, 2, 3], indicating its validity. Tables 3 and 8 illustrate the results, which show that only a little privacy was revealed.
>
> [a] When Machine Unlearning Jeopardizes Privacy
>
> ```
> Q3. Input encryption techniques [4,5,6] are widely used to protect sensitive data during model training. Could you briefly compare how graph unlearning differs from input encryption in terms of privacy guarantees and practical deployment?
> ```
>
> Q3: This is a very interesting question. We investigated the input encryption techniques and summarized the following:
>
> **Practical Deployment:**
>
> 1. **Deployment Purpose:** Input encryption is a method to effectively mitigate privacy risks without slowing down training or reducing accuracy in distributed or federated learning; graph unlearning aims to effectively remove the influence of nodes or edges without retraining.
> 2. **Deployment Order:** Input encryption plays a role only in data preparation before model training in a similar way as mixup training; our graph unlearning takes effect after model learning, while there are some method achieves efficient unlearning by regularizing model training.
>
> **Privacy Guarantees:**
>
> 1. Input encrytion ensure any attack must solves a computationally expensive task for privacy guarantee.
> 2. Unlearning seeks to reduce the differences in output of retraining and unlearning for indistinguishability of data.

---

> ### Author Response · Authors · 2025-11-24
>
> Dear Reviewer htoX,
>
> As the rebuttal discussion phase is coming to an end, we would like to know if our response has addressed your concerns and questions. If you have any remaining feedback that could help raise the rating, we would greatly appreciate it.
>
> Thank you again for the time and effort you have dedicated to reviewing our work.
>
> Best regards,
>
> Authors

---

### Official Review · Reviewer_dj7s · 2025-11-01

**Soundness:** 2
**Presentation:** 2
**Contribution:** 2
**Rating:** 4
**Confidence:** 4

**Summary:**

This paper proposes an efficient graph unlearning method that avoids expensive retraining. Instead of updating full GNN parameters, the authors reverse GNN message passing: they decompose the influence of an unlearned node, reconstruct its embedding from neighbors, and subtract this influence from retained nodes. A range–null space decomposition is introduced to correct reconstruction errors with theoretical guarantees. The framework achieves up to 40× faster unlearning than retraining while maintaining comparable model utility and privacy across various GNN architectures and datasets.

**Strengths:**

1. The approach does not depend on a specific GNN architecture.
2. The method significantly reduces computational cost.
3. Experimental results show that the method achieves effective forgetting of target nodes while keeping performance on retained nodes close to the retrained-from-scratch baseline.

**Weaknesses:**

1. Although key methods like GraphEraser, GIF, and GNNDelete are included, the paper does not compare against very recent graph unlearning methods. This makes it unclear how competitive the method is versus the latest developments.
2. The paper emphasizes low FLOPs overhead (~0.3%), but it only reports theoretical FLOPs, not actual runtime, or wall-clock training time on large graphs or large GNNs. No analysis is provided on how the method scales to industrial graph sizes or deeper GNNs.
3. This paper suffers from inconsistent citation and formatting styles that do not fully follow standard English academic writing norms—for example, improper spacing after mathematical symbols and duplicated content such as the repetition around line 104.

**Questions:**

Figure 4 does not include comparisons with any baseline methods, making it difficult to evaluate the unlearning efficacy of the proposed approach relative to existing techniques. In addition, the framework introduces several new components (interaction modeling, reconstruction, RND rectification, local refinement), but no pseudocode or algorithm is provided. This makes reproducibility unclear, and the added complexity raises concerns about computational overhead—what is the actual time/memory cost, and how do you justify that this added burden is worthwhile in practice?
Moreover, the challenges stated in the Introduction do not seem to be fully aligned with what the method actually solves, and the analysis is somewhat imprecise. In the multiple unlearning request scenario, how does your method perform compared to baselines? What are the empirical results for your method under multiple sequential unlearning requests, and how stable or scalable is it in that setting?

---

> ### Author Response · Authors · 2025-11-14
>
> ```
> W1. Although key methods like GraphEraser, GIF, and GNNDelete are included, the paper does not compare against very recent graph unlearning methods.
> ```
>
> W1: We appreciate this concern. Due to the limitation of the main body, more comparisons are presented in the appendix.
> 1. In fact, our appendix (Table 4 in Section B) already includes comparisons with MEGU, IDEA, ScaleGUN, GraphRevoker, and other recent methods from 2024 to 2025 on the SGC backbone. The results show our method achieves competitive or superior performance.
> 2. More comprehensive comparisons are also shown as Tables 5,6, and 7 under the backbone of GAT, GCN, and GIN. Representative methods that perform well in Table 4 are listed in these tables, including the latest methods.
> 3. Moreover, Appendix B.1 contains discussions of the properties and performance of these methods.
>
> ```
> W2. The paper emphasizes low FLOPs overhead (~0.3%), but it only reports theoretical FLOPs, not actual runtime, or wall-clock training time on large graphs or large GNNs. No analysis is provided on how the method scales to industrial graph sizes or deeper GNNs.
> ```
>
> W2: We appreciate this important concern. The theoretical FLOPs analysis (0.3% overhead) is validated by our runtime measurements showing 40-88× speedup over retraining.
> 1. Our current experiments report both actual running time (RT) in seconds (Tables 2, 4, 5, 6, 7) and theoretical FLOPs (Tables 9, 10, 11, 12).
> 2. Since the low extra FLOPs, our method could be easily applied to industrial graph. Noting that the linear complexity makes it particularly suitable for large-scale deployment.
> 3. Reviewer's consideration about deeper GNNs is very insightful. Due to the page limitation, the discussion has already been presented in Appendix D.1. Hope this addresses your concerns.
>
> ```
> W3. This paper suffers from inconsistent citation and formatting styles that do not fully follow standard English academic writing norms.
> ```
>
> W3: We apologize for these issues and will address them comprehensively in revision. Duplicated 'Guo et al.' around line 104 would be fixed. We will systematically review and correct all citation formats.
>
> ```
> Q1. Figure 4 does not include comparisons with any baseline methods.
> ```
>
> Q1: Thanks for your thorough review. Figure 4 currently shows only our method, retraining, and the noisy baseline to measure the effectiveness of our method in mitigating the negative impact of adversarial data. However, most papers have not covered efficacy, so we presented only a comparison of efficiency, utility, and privacy in the submission. Additional comparisons will be provided in the revised Appendix G. It is worth noting that only our method achieves an improvement over `Noisy' in all backbones and datasets.
>
> ```
> Q2. No pseudocode or algorithm is provided.
> ```
>
> Q2: We agree that pseudocode is essential for reproducibility. We will add a comprehensive algorithm/pseudocode section in Appendix E in the revision. It could also refer to the submitted code for reproduction.
>
> ```
> Q3. How do you justify that this added burden is worthwhile in practice?
> ```
>
> Q3: Thanks for your insightful concerns. It is noted that the costs we listed in Tables 9, 10, 11, and 12 are in the form of relative percentages compared with the backbone. An extra 0.3% FLOPs and 2 MLP modules are negligible under the background of contemporary computing power, thus worthwhile.
>
> ```
> Q4. Moreover, the challenges stated in the Introduction do not seem to be fully aligned with what the method actually solves.
> ```
>
> Q4: We appreciate this feedback. We would give a more concise and clear correspondence:
>
> 1. **Challenge 1: Complicated entanglement of node features:** We model node-wise interactions through $f_1$, which explicitly captures how unlearned nodes influence their neighbors. The reconstruction process ($f_2$) reverses this influence, allowing us to disassemble the entanglement.
>
> 2. **Challenge 2: Repetitive computation overhead for multiple requests:** By modifying output embeddings rather than parameters full fine-tuning, we can handle multiple unlearning requests on shared embeddings without retraining. The interaction model is composed of only two MLPs, which are trained with very low cost.
>
> ```
> Q5. Multiple unlearning request performances compared to baselines? Empirical results under multiple sequential unlearning requests, and how stable or scalable is it?
> ```
>
> Q5: Thanks for your professional concerns. We illustrate sequential unlearning through Figure 6. While the main text presents these as varying unlearning ratios (10%, 20%, 30%, 40%), the implementation is actually sequential unlearning, demonstrating our method's capability for multiple sequential requests. Our method maintains utility close to retraining while being computationally efficient. Compared with unlearning method based on influence function like GIF, there is an obvious advantage. And more comparison and detailed description would be presented in revision.

---

> ### Author Response · Authors · 2025-11-24
>
> Dear Reviewer dj7s,
>
> As the rebuttal discussion phase is coming to an end, we would like to know if our response has addressed your concerns and questions. If you have any remaining feedback that could help raise the rating, we would greatly appreciate it.
>
> Thank you again for the time and effort you have dedicated to reviewing our work.
>
> Best regards,
>
> Authors

---

### Official Review · Reviewer_CBm5 · 2025-11-01

**Soundness:** 3
**Presentation:** 2
**Contribution:** 3
**Rating:** 4
**Confidence:** 3

**Summary:**

The paper addresses the problem of node-unlearning in GNNs. The authors propose to reverse the aggregation process by modelling node-wise interactions. To handle the fact that embedding transformations are often irreversible, they employ a Range-null space decomposition (RND) framework. Their approach modifies only embeddings and keep embedding distributions similar before and after unlearning. Experimental results on several real-world graph datasets demonstrate significant speed-ups and comparable model utility and privacy robustness.

**Strengths:**

S1. The experimental evaluation is thorough and comprehensive, covering multiple benchmark datasets and GNN architectures to demonstrate both efficiency and utility.

S2. The method design is intuitive and interpretable, the authors provide a clear workflow for node unlearning.

S3. Transferring RND decomposition into graph-unlearning context is also intuitive for addressing challenges of inverse reconstruction.

S4. Theoretical analysis is also provided with framework design.

**Weaknesses:**

(Most Important) W1. Lack the section of “The Use of Large Language Models (LLMs)”. As the organizer stated: “Not disclosing significant LLM usage can lead to desk rejection of the paper.”

W2. The method’s reliance on the pseudo-inverse of the layer Jacobian matrix
𝐻
H

assumes numerical stability and invertibility, but the paper does not sufficiently analyse or mitigate potential ill-conditioning of
𝐻
H

.

W3. The learnable component f_2 is presumed to capture the null-space residuals of the range–null decomposition, yet the paper does not provide rigorous guarantees or regularization to ensure that f_2  does not overfit noise.

W4. The approach focuses on embedding modification, which raises the concern that if the removed node had a significant gradient influence on the model parameters (e.g., centrality node), which may not sufficiently erase its impact at the parameter level.

W5. While the RND technique is novel in the context of graph unlearning, the underlying mathematical tool is established in other ML/signal tasks, especially in recent two years [1],[2],[3], so the novelty of method design is somewhat incremental.

W6. Many spelling, grammar, and citation typos are found:

Line 32-33: “exhorbitant computational overhead”.

Line 275-276: “Although we does not…, but …”.

Line 882-883: “Then, we prove that … is projecton matrix”.

Line 275-276: “Although …, but …”

In addition, the majority of wrong citations usage, i.e. ~\cite{} rather than ~\citep{}, this constitutes a format inconsistency.

---

References

[1] Wang, Yinhuai, et al. "Gan prior based null-space learning for consistent super-resolution." Proceedings of the AAAI Conference on Artificial Intelligence. Vol. 37. No. 3. 2023.

[2] Chen, Jiacheng, et al. "Null space matters: range-null decomposition for consistent multi-contrast MRI reconstruction." Proceedings of the AAAI Conference on Artificial Intelligence. Vol. 38. No. 2. 2024.

[3] Li, Andong, et al. "Learning Neural Vocoder from Range-Null Space Decomposition." arXiv preprint arXiv:2507.20731 (2025).

**Questions:**

Please check the weaknesses part

---

> ### Author Response · Authors · 2025-11-14
>
> ```
> (Most Important) W1. Lack the section of “The Use of Large Language Models (LLMs)”. As the organizer stated: “Not disclosing significant LLM usage can lead to desk rejection of the paper.”
> ```
>
> W1: We sincerely apologize for this oversight. We acknowledge that we did not use any Large Language Models (LLMs) in the preparation of this manuscript. All writing, analysis, and experimental work were conducted by the authors without LLM assistance. We will add a section titled "The Use of Large Language Models (LLMs)" to explicitly state that no LLMs were used in this work, in compliance with ICLR 2026 submission guidelines. It would be presented in appendix F.
>
> ```
> W2. The method’s reliance on the pseudo-inverse of the layer Jacobian matrix 𝐻 H assumes numerical stability and invertibility, but the paper does not sufficiently analyse or mitigate potential ill-conditioning of 𝐻 H.
> ```
>
> W2: We appreciate this important concern. The reviewer is correct that we should address potential ill-conditioning of matrix $H$. In our implementation, we use the Moore-Penrose pseudo-inverse $H^{\dagger}$ which is well-defined even for singular or ill-conditioned matrices. The pseudo-inverse provides a least-squares solution and is numerically stable through standard linear algebra libraries (e.g., numpy.linalg.pinv).
>
> ```
> W3. The learnable component f_2 is presumed to capture the null-space residuals of the range–null decomposition, yet the paper does not provide rigorous guarantees or regularization to ensure that f_2 does not overfit noise.
> ```
>
> W3: This is a valid concern. The loss $L^\text{Local}$ uses KL divergence between normalized embeddings, which inherently provides regularization for $f_1$ so as $f_2=MLP_2(f_1)$, the function of $f_1$. Additionally, $f_2$ is trained by $L^\text{Inter}$ to reconstruct $(k-1)$-th layer embeddings, not arbitrary noise. The theoretical guarantee in Theorem 1 shows that $f_2^\text{RND}$ minimizes the $L_2$ error compared to the optimal $f_2^*$, which also bounds the overfitting risk of $f_2$.
>
> ```
> W4. The approach focuses on embedding modification, which raises the concern that if the removed node had a significant gradient influence on the model parameters (e.g., centrality node), which may not sufficiently erase its impact at the parameter level.
> ```
>
> W4: This is an insightful question about the fundamental design choice. Our method modifies embeddings rather than parameters. For a fixed GNN architecture, the mapping from input to output embeddings is deterministic. We train MLPs that modeling node-wise interactions to match those of a retrained model equivalently.
>
> For high-centrality nodes with significant gradient influence, our method handles similar problem through unified design:
> 1. The $\gamma$ scaling factor in Eq. (9) would ajust for high influenced nodes, which is positively correlated with average degree of unlearned nodes.
> 2. Embeddings switch from $h^k$ to $\overline{h}^k$ with inference on modified graph $G'$ for significant graph structure change in appendix C.2, where unlearned centrality node take the same effect of graph structure change.
>
> ```
> W5. While the RND technique is novel in the context of graph unlearning, the underlying mathematical tool is established in other ML/signal tasks, especially in recent two years [1],[2],[3], so the novelty of method design is somewhat incremental.
> ```
>
> W5: We appreciate the reviewer's concern about novelty. While RND has been used in other domains, we provides a reference for researchers who wanna try and explore similar problem like graph unlearning. Besides, our key contributions also includes:
>
> 1. **Application domain adaptation**: We are the first to apply RND to graph unlearning, which presents unique challenges: $\textbf{(1)}$ Graph structure involves non-Euclidean dependencies between nodes, where we must additionally model node-wise interactions rather than pixel/voxel-level operations. $\textbf{(2)}$ The reverse aggregation problem (identifying the target of reconstructing $(k-1)$-th layer from $k$-th layer) is fundamentally different from image/video reconstruction tasks.
> 2. **Theoretical framework**: We establish the linear inverse constraint between $f_1$ and $f_2$ specifically for GNN message passing, proving that $f_2^\text{RND}$ has smaller error than initial estimation (Theorem 1), which is tailored to the graph unlearning setting.
>
> ```
> W6. Many spelling, grammar, and citation typos are found
> ```
>
> W6: We sincerely apologize for these typos and will correct them in the revision:
>
> 1. Line 32-33: "exhorbitant" $\rightarrow$ "exorbitant"
>
> 2. Line 275-276: removing the redundant "but"
>
> 3. Line 882-883: "projecton" $\rightarrow$ "projection"
>
> 4. We will systematically manage \cite{} and \citep{} throughout the manuscript to ensure consistent citation formatting.
>
> We will conduct a thorough proofreading pass to fix all spelling, grammar, and formatting issues in revision.

---

> ### Author Response · Authors · 2025-11-24
>
> Dear Reviewer CBm5,
>
> As the rebuttal discussion phase is coming to an end, we would like to know if our response has addressed your concerns and questions. If you have any remaining feedback that could help raise the rating, we would greatly appreciate it.
>
> Thank you again for the time and effort you have dedicated to reviewing our work.
>
> Best regards,
>
> Authors

---

### Author Response · Authors · 2025-11-15
**Rebuttal Revision Submission Version 1.0**

Dear Reviewers,

We appreciate your questions and suggestions. This revision (Revision 1.0) addresses the writing-related concerns you raised. We have corrected spelling and grammar errors throughout the manuscript. We have also included algorithm flowcharts and LLM usage in the appendix. All changes in the revised manuscript are highlighted in blue for your convenience. Additional experiments will be included in subsequent revision versions.

We have also provided detailed responses to your reviews, addressing each point. Hope these address your concerns.

Best regards,

Authors

---

### Author Response · Authors · 2025-11-15
**Rebuttal Revision Submission Version 2.0**

Dear Reviewers,

This revision (Revision 2.0) provides additional experiments concerning efficacy comparison in Appendix G. The corresponding experimental analysis is also attached. Including all previous modifications, are still marked in blue.

Hope to get your feedback.

Best regards,

Authors

---

### Author Response · Authors · 2025-12-02
**Author Final Remarks**

Dear AC:

We thank all reviewers for their constructive feedback. In this response, we address the main concerns raised by reviewers CBm5, dj7s, and htoX. We have organized concerns and our responses into three categories:

- $\text{(1) Writing and formatting issues};$
- $\text{(2) Methodological and theoretical concerns};$
- $\text{(3) Experimental evaluation concerns.}$

All identified issues have been addressed in our revision, and we clarify several factual misunderstandings in the reviewers' comments.

**Writing and Formatting Issues.** We provided a clear workflow for our method (CBm5) and fixed some writing and citation issues as:

- $\text{F1. Missing LLM usage disclosure section}$
- $\text{F2. Citation formatting inconsistencies}$

All issues have been corrected in revision. See rebuttal: **CBm5 W1, W6; dj7s W3**.

**Methodological and Theoretical Concerns.** The reviewers acknowledged the effectiveness of our method for interpretability in theoretical (CBm5, htoX) and model-agnostic approaches (dj7s). We further clarified:

- $\text{M1. Numerical stability of matrix H}$
- $\text{M2. Overfitting risk of} f_2$
- $\text{M3. Theoretical justification for} (k-1)\text{-th layer}$
- $\text{M4. Case-specific formulations}$

$\text{[M1]:}$ We use Moore-Penrose pseudo-inverse $H^{\dagger}$, which is well-defined for singular/ill-conditioned matrices and provides numerical stability. See rebuttal: **CBm5 W2**.

$\text{[M2]:}$ Theorem 1 already provides theoretical bounds showing $f_2^{\text{RND}}$ minimizes $L_2$ error, bounding overfitting risk. See rebuttal: **CBm5 W3**.

$\text{[M3]:}$ We provide three principled reasons for $(k-1)$-th layer choice. See rebuttal: **htoX W1**.

$\text{[M4]:}$ Reviewer characterizes extensions as "ad-hoc modifications." Our method employs principled case-specific adaptations that are unified within a general framework applicable to all situations. See rebuttal: **htoX W2**.

**Experimental Evaluation Concerns.** Reviewer CBm5 and htoX agrees that the experimental evaluation is thorough and comprehensive. We clarify the reviewer dj7s's factual misunderstandings as:

- $\text{E1. Comparison with recent methods}$
- $\text{E2. Runtime vs. theoretical FLOPs}$
- $\text{E3. Efficacy evaluation}$
- $\text{E4. Pseudocode and reproducibility}$
- $\text{E5. Computational overhead justification}$
- $\text{E6. Sequential unlearning}$

$\text{[E1]:}$ We respectfully clarify a factual misunderstanding "does not compare against very recent graph unlearning methods." Appendix Tables 4-7 already include comparisons with MEGU, IDEA, ScaleGUN, GraphRevoker (2024-2025) across multiple backbones. See rebuttal: **dj7s W1**.

$\text{[E2]:}$ We respectfully clarify factual misunderstandings "only reports theoretical FLOPs, not actual runtime." and "No analysis on industrial-scale graphs or deeper GNNs." Tables 2, 4-7 already report actual running time (RT) in seconds, showing 40-88× speedup. Discussion on deeper GNNs is in Appendix D.1. See rebuttal: **dj7s W2**.

$\text{[E3]:}$ We added baseline comparisons in revision. See rebuttal: **dj7s Q1**.

$\text{[E4]:}$ Added comprehensive algorithm pseudocode in Appendix E. See rebuttal: **dj7s Q2**.

$\text{[E5]:}$ Added complexity (0.3% FLOPs, 2 MLPs) is negligible; 40-88× speedup justifies it. See rebuttal: **dj7s Q3**.

$\text{[E6]:}$ Figure 6 already demonstrates sequential unlearning (ratios 10%-40% are implemented sequentially). See rebuttal: **dj7s Q5**.

We have effectively addressed all reviewers' concerns and have incorporated more comprehensive analyses in our revision. We respectfully request the Area Chair's consideration of our responses and the improvements made in the revision.

Best Wishes,

Authors

---

### Meta-Review · Area_Chair_L2Ri · 2026-01-09

**Summary:**

The paper presents an efficient graph node unlearning method that reverses GNN message passing via embedding reconstruction with range–null space decomposition, achieving substantial speedups over retraining while largely preserving model utility and privacy. Despite authors' comprehensive rebuttal addressing points and revisions incorporating additional experiments, reviewers maintained concerns about the informal justification of design choices and the need for case-specific formulations, leading to the rejection decision.

**Reviewer Concerns:**

Reviewers consistently praised the method’s intuitive design, model-agnostic nature, solid theoretical grounding, and strong empirical results across multiple datasets and GNN backbones. However, there was also consensus on several weaknesses that made the consensus to reject the paper: the justification for key design choices remains somewhat informal (despite the fact authors tried to resolve in rebuttal and additional results in revised version of paper), the need for case-specific extensions weakens the perceived generality of the framework, and presentation issues (clarity, formatting, and reproducibility details). Overall, the paper is well grounded, and addressing these issues would further strengthen it.

**Reviewer Scores:**

Reviewer CBm5: concerns on computational issues (SVD to compute pseudo-inverse) + better articulation of novelty on theoretical side\
Reviewer dj7s: through comparison with other methods + improving presentation\
Reviewer htoX: making the design choices more clear and grounded + discussion on generality of proposed method\

---

### Decision · Program_Chairs · 2026-01-26

Reject